# Modeled Greenland Ice Sheet evolution constrained by ice-core-derived Holocene elevation histories

Mikkel L. Lauritzen[1], Anne Solgaard[3], Nicholas Rathmann[1], Bo M. Vinther[1], Aslak Grindsted[1], Brice Noël[4], Guðfinna Aðalgeirsdóttir[2], and Christine S. Hvidberg[1]

[1]Niels Bohr Institute, University of Copenhagen, Copenhagen, Denmark
[2]Institute of Earth Sciences, University of Iceland, Reykjavík, Iceland
[3]Geological Survey of Denmark and Greenland, Copenhagen, Denmark
[4]Laboratoire de Climatologie et Topoclimatologie, SPHERES, University of Liège, Liège, Belgium

**Correspondence:** Mikkel L. Lauritzen (mikkel.lauritzen@nbi.ku.dk)

**Abstract.** During the Holocene, the Greenland Ice Sheet (GrIS) experienced substantial thinning, with some regions losing up to 600 meters of ice. Ice-sheet reconstructions, paleoclimatic records, and geological evidence indicate that during the Last Glacial Maximum, the GrIS extended far beyond its current boundaries and was connected with the Innuitian Ice Sheet (IIS) in the northwest. We investigate these long-term geometry changes and explore several possible factors driving those changes by using the Parallel Ice Sheet Model (PISM) to simulate the GrIS thinning throughout the Holocene period, from 11.7 ka ago to the present. We perform an ensemble study of 841 model simulations in which key model parameters are systematically varied to determine the parameter values that, with quantified uncertainties, best reproduce the 11.7 ka of surface elevation records derived from ice cores, providing confidence in the modeled GrIS paleo evolution. We find that since the Holocene onset, 11.7 ka ago, the GrIS mass loss has contributed 5.3±0.3 m to the mean global sea-level rise, which is consistent with the ice-core-derived thinning curves spanning the time when the GrIS and the Innuitian Ice Sheet were bridged. Our results suggest that the ice bridge collapsed 4.9±0.5 ka ago and that the GrIS is still responding to these past changes, having raised sea level by 23±26 mm SLE ka$^{-1}$ in the last 500 years. Our results have implications for future ice-sheet evolution, which should account for this long-term, transient trend.

## 1 Introduction

During the Last Glacial Maximum (LGM), approximately 20 ka ago, Earth was covered by large ice sheets, including the Laurentide, Fenno-Scandian, Innuitian, and Greenlandic ice sheets, and the global mean sea level was 125–134 m lower than today (Lambeck et al., 2014; Yokoyama et al., 2018). Geological evidence suggests that the Greenland Ice Sheet (GrIS) extended to the continental shelf and was connected to the Innuitian Ice Sheet (IIS) at the Nares Strait (England et al., 2006).

Towards the end of the last glacial period, the Bølling-Allerød interstadial brought abrupt warming to the Northern Hemisphere 14.7 ka ago, followed by the cooling of the Younger Dryas stadial 12.9 ka ago (Rasmussen et al., 2006). The Holocene interglacial began 11.7 ka ago, bringing temperatures that were locally up to 15°C warmer in Greenland (Andersen et al., 2004). However, temperature reconstructions vary by several degrees, which is crucial for the GrIS Holocene evolution (Nielsen et al.,

2018). Following the Holocene Thermal Maximum, 6-9 ka ago, Greenland temperatures have shown a long-term decreasing trend (Vinther et al., 2009), but anthropogenic forcing has since reversed the course of natural temperature change, resulting in a global increase in temperatures since pre-industrial times (Eyring et al., 2021).

Accurately modeling the historical evolution of the GrIS is essential for evaluating and calibrating ice-sheet models. Ice-sheet models respond to climate change over a range of timescales and are rarely in steady state (e.g. Lauritzen et al., 2023). However, several ice-sheet model studies have overlooked a calibration of their temporal evolution and only focused on the evolution of ice temperature, neglecting other delayed responses, such as bedrock dynamics. For example, the ISMIP6 protocol does not require calibration (Nowicki et al., 2020), and most of the ISMIP6 ensemble simulations underestimate the observed IMBIE consensus mass loss from the GrIS (The IMBIE Team, 2020; Aschwanden et al., 2021). Recent advances have addressed this by calibrating an ice-sheet model to satellite-based gravimetry-derived mass loss data of the GrIS (Aschwanden and Brinkerhoff, 2022), but the satellite-based calibration data period only covers 22 years at the time of writing, and there is no guarantee that it gives a sensible long-term response.

Calibrating the model to align with present-day observations of ice thickness and velocities risks capturing only the present-day state while being on a wrong state trajectory; that is, neglecting the long-term memory of the ice sheet and the response of the bedrock to past changes in ice load. These differences in past trajectories affect the projected future mass loss in this century as demonstrated by Aðalgeirsdóttir et al. (2014).

To simulate time periods before the satellite era, ice-sheet modeling must rely on proxy data from paleo-climatic records and ice extent markers for constraining and validating the long-term transient response of the ice sheet (state trajectory) over these considerably longer timescales.

Past temperatures can be inferred from oxygen isotope measurements. When water evaporates from the oceans and precipitates over the GrIS, a temperature-dependent fractionation process alters the ratio of oxygen isotopes in the water — a relationship first used by Dansgaard et al. (1969) to infer past temperatures from oxygen isotope measurements at Camp Century (CC). Vinther et al. (2009) used this temperature dependence to derive a GrIS-wide oxygen isotope signal by assuming that the Renland and Agassiz (see Fig. 1) ice-core sites are located within restricted ice domes where ice thickness remains constant. This GrIS-wide oxygen isotope signal was then subtracted from the oxygen isotope signals at CC, NGRIP, GRIP, and Dye 3 (see Fig. 1) to derive local surface elevation histories, after correcting for upstream effects. These surface elevation histories provide constraints for modeling the GrIS throughout the Holocene, offering valuable insights into the ice sheet's response to past climate changes and helping to improve the robustness of model predictions.

Previous studies have attempted to model elevation changes derived from ice cores. Notably, Lecavalier et al. (2017) modeled Holocene surface elevation changes at CC using temperature anomalies from the Agassiz ice cores, suggesting that early Holocene temperatures were 7°C higher than today. However, they did not account for the buttressing effect of the IIS, a key driver of thinning (MacGregor et al., 2016), and focused only on relative elevation changes, without reconstructing absolute elevation history. More recently, Tabone et al. (2024) successfully modeled elevation changes at the GRIP site, attributing them to the onset of the Northeast Greenland Ice Stream (NEGIS).

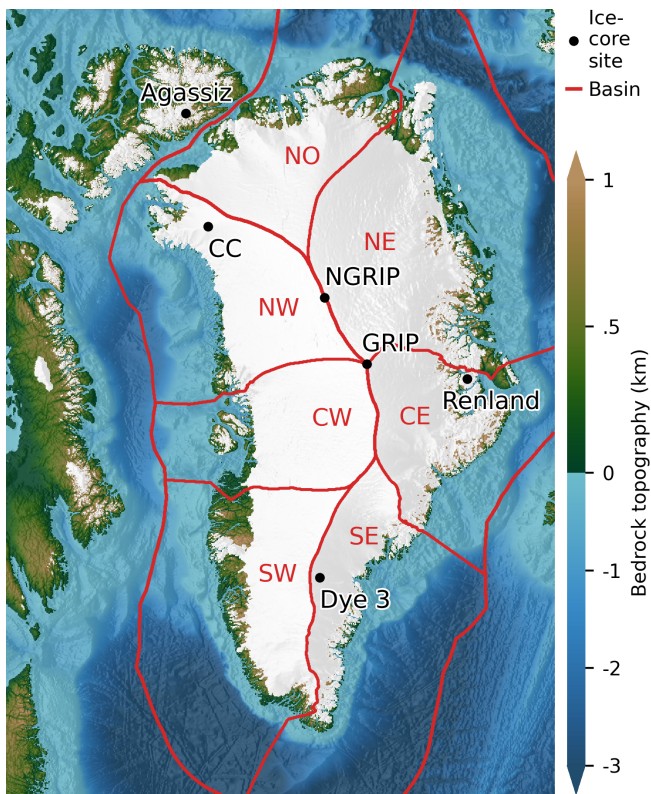

**Figure 1.** Model domain showing the present-day bedrock topography from Morlighem (2022), Jakobsson et al. (2020), and GEBCO Bathymetric Compilation Group (2023) with the present-day ice cover from Morlighem (2022) and RGI Consortium (2023) shown in white. The ice-core sites discussed in the text (CC, NGRIP, GRIP, Dye 3, Renland, and Agassiz) are shown on top together with the glacier catchment basins (NW, CW, SW, SE, CE, NE, NO) from Mouginot and Rignot (2019) and extended out to the Exclusive Economic Zone of Greenland (Flanders Marine Institute (VLIZ), Belgium, 2023) and constitute our extended continental shelf (ECS) domain, see text.

In this study, we use the Parallel Ice Sheet Model (PISM) to model the long-term transient response of the GrIS to past climatic changes and the collapse of the IIS bridge during the Holocene. By varying 20 influential model parameters in an ensemble of 841 members, we show that it is possible to model the ice-core-derived elevation histories rather than just the thinning *if* the grounding line can advance to the continental shelf *and* the GrIS can connect to the IIS. We use this setup to constrain the model parameters for the ensemble and to estimate the GrIS long-term evolution with quantified uncertainties. Using the calibrated model, we investigate the Holocene ice sheet mass loss and assess the ongoing long-term response of the modeled GrIS and bedrock dynamics.

## 2  Model setup

To model the Holocene evolution of the GrIS, we use the open-source Parallel Ice Sheet Model (PISM) version 2.1 (Bueler and Brown, 2009; Winkelmann et al., 2011) at 20 km resolution for the spin-up, refined to 10 km for the last 20 ka. PISM is a three-dimensional thermomechanically-coupled model that solves both the Shallow Ice Approximation (SIA) and Shallow Shelf Approximation (SSA) in a hybrid scheme, capturing both slow-moving interior flow and fast flow in ice streams and outlet glaciers. At the ice-ocean boundary, PISM includes sub-grid parameterizations to model grounding line advance and retreat (Gladstone et al., 2010). Model parameters, listed in Table 1, will be varied in our ensemble simulations unless otherwise specified.

### 2.1  Model domain

The model domain, shown in Fig. 1, spans $6.7 \times 10^6$ km$^2$, from the continental shelf in the east to the Canadian Arctic Archipelago in the west. A north polar stereographic projection with a standard parallel at 70° N and central longitude of -45° W (ESPG 3413) is used. The projection introduces distortions of up to +5% in the north and -11% in the southwest relative to the central latitude and longitude. PISM uses a flat earth approximation, so volume is not conserved when transforming thickness between projections. Volumes and mass loss rates are reported using the actual grid area.

To partition mass between Greenland and Canada, we introduce an Extended Continental Shelf (ECS) mask, corresponding to Greenland's Exclusive Economic Zone (Flanders Marine Institute (VLIZ), Belgium, 2023). This divides the two regions at the Nares Strait and Baffin Bay, extending to the continental shelf in the north, east, and south. For mass loss partitioning when the grounding line advances, we extend the basins from Mouginot and Rignot (2019) to the ECS using nearest neighbor extrapolation.

The present-day bedrock topography over Greenland is from BedMachine v5 (Morlighem, 2022), and extended with IBCAO v4.2 (Jakobsson et al., 2020) and GEBCO Bathymetric Compilation Group (2023) to cover the larger domain, in that order of preference to get the best bedrock available.

At the lateral boundary, a Dirichlet boundary condition of zero ice thickness is used, and the influence of the majority of the Laurentide Ice Sheet is thereby neglected. The north and south are bounded by the open ocean, while Iceland and Svalbard are just visible towards the east. At the base of a 2 km deep bedrock thermal layer, the thermal heat flux from Shapiro (2004) is applied constantly in time.

### 2.2  Atmospheric forcing

To model the surface mass balance (SMB), we apply a Positive Degree Day (PDD) scheme to calculate the surface melting. This approach bases the SMB solely on temperature, $T$, and precipitation, $P$. In the PDD scheme, surface melt rate is proportional to the extent to which the temperature exceeds the freezing point (e.g. Braithwaite, 1985),

$$\dot{m}^s \propto \max(0°\text{C}, T), \tag{1}$$

where we use two constants of proportionality: one for snow, $f_s$, and another for ice, $f_i$.

To force the PDD model, we use a 12-month reference climatology based on the multi-year monthly averages of temperature and precipitation for the period 1960–1989 from RACMO (Noël et al., 2015, 2018, 2019). Since our model domain is not covered by a single RACMO simulation, we combine different simulations (see Fig. A3). We merged three areas with precipitation data: Greenland, the Northern Canadian Arctic Archipelago, and the Southern Canadian Arctic Archipelago (Noël et al., 2018), treating areas outside these regions as having no precipitation. For temperature, we used RACMO2.3p2 at 5.5 km for Greenland (Noël et al., 2019) and combined it with a broader 11 km simulation (Noël et al., 2015) for the rest of the area. The mean precipitation and summer temperatures from the resulting climatology are shown in Fig. A2.

Following Nielsen et al. (2018), we account for paleo temperature changes by applying a spatially uniform, time-varying temperature anomaly, $\Delta T$, along with a lapse rate adjustment, $\Gamma$, which modifies the surface temperature based on deviations from the RACMO surface topography. The temperature reconstructions are shown in Fig. A1. Insolation changes are not included in this approach.

Reconstructions 1 and 2 use the GRIP ice core with linear and quadratic transfer functions from Huybrechts (2002) and Johnsen et al. (1995), respectively. Reconstruction 3 uses the NGRIP core with the same transfer function from Huybrechts (2002), while Reconstruction 4 is the GrIS-wide reconstruction from Vinther et al. (2009) — the only one that accounts for elevation change. Reconstruction 5 is based on the NGRIP core with an isotope diffusion inversion scheme (Gkinis et al., 2014).

The Holocene Thermal Maximum is only captured by Reconstructions 3, 4, and 5, while Reconstructions 1 and 2 suggest a more constant Holocene climate. Reconstruction 1 was used by the SeaRISE project (Bindschadler et al., 2013).

Since the vapor pressure scales approximately exponentially with temperature in the Clausius-Clapeyron relation, we account for paleo precipitation changes by scaling the reference precipitation field with a time-dependent scaling factor $\exp(\omega(\phi)\Delta T(t))$. Here, $\omega$ has the latitude dependence

$$
\omega(\phi(x,y)) = \begin{cases} \omega_\downarrow & \phi \leq \phi_\downarrow^p \\ \omega_\downarrow + \dfrac{\phi - \phi_\downarrow^p}{\phi_\uparrow^p - \phi_\downarrow^p}(\omega_\uparrow - \omega_\downarrow) & \phi_\downarrow^p \leq \phi \leq \phi_\uparrow^p , \\ \omega_\uparrow & \phi_\uparrow^p \leq \phi \end{cases}
\tag{2}
$$

where $\phi$ is the latitude and $\phi_\downarrow^p = 60°N$ and $\phi_\uparrow^p = 75°N$ are chosen to cover most of Greenland. $\omega_\downarrow$ and $\omega_\uparrow$ are the southern and northern precipitation scaling parameters, respectively, which will be varied in our ensemble. This approach allows for different precipitation histories in Northern and Southern Greenland, in contrast to the uniform scaling used in many previous modeling attempts (e.g. Nielsen et al., 2018).

## 2.3 Ocean forcing

Following Aschwanden et al. (2019) we take the sub-shelf ocean melt to be separable in space and time

$$
\dot{m}^o(x,y,t) = \dot{m}_x^o(\phi(x,y))\dot{m}_t^o(t),
\tag{3}
$$

with the spatial dependence controlling the present-day melt rate given by

$$
\dot{m}_x^o(\phi(x,y)) = \begin{cases} \dot{m}_\downarrow^o & \phi \leq \phi_\downarrow^o \\ \dot{m}_\downarrow^o + \dfrac{\phi - \phi_\downarrow^o}{\phi_\uparrow^o - \phi_\downarrow^o}(\dot{m}_\uparrow^o - \dot{m}_\downarrow^o) & \phi_\downarrow^o \leq \phi \leq \phi_\uparrow^o \\ \dot{m}_\uparrow^o & \phi_\uparrow^o \leq \phi \end{cases}, \tag{4}
$$

where $\phi_\downarrow^o = 71°N$ and $\phi_\uparrow^o = 80°N$, following Aschwanden et al. (2019), while $\dot{m}_\downarrow^o$ and $\dot{m}_\uparrow^o$ are the upper and lower melt values which we will vary. To allow the formation of an ice bridge to Canada, the sub-shelf melt rate is scaled by

$$
\dot{m}_t^o(t) = \begin{cases} 0 & t \leq \tau \\ \dfrac{t - \tau}{\Delta\tau} & \tau \leq t \leq \tau + \Delta\tau \\ 1 & \tau + \Delta\tau \leq t \end{cases}, \tag{5}
$$

such that there will be no ocean melt for times earlier than $\tau$, while it increases to present-day values in the time $\Delta\tau$ inspired by the rapid change in ocean temperatures found by Clark et al. (2020). In addition to sub-surface melt, ice is calved off at the oceanfront at a rate that is proportional to the tensile von Mises stress and inversely proportional to a characteristic parameter, $\sigma_{\mathrm{max}}$ (Morlighem et al., 2016). Additionally, all ice thinner than $H_{\mathrm{cr}}$ is calved off, and a eustatic sea level forcing from Imbrie and McIntyre (2006) is applied, changing the ocean level by 130 m in the last 19 ka.

## 2.4 Ice dynamics

The constitutive relation that relates the strain rate, $\dot{\epsilon}_{ij}$, to the stress, $\tau_{ij}$, in the ice sheet is

$$
\dot{\epsilon}_{ij} = EA\tau_{\mathrm{e}}^{n-1}\tau_{ij}, \tag{6}
$$

where $E$ is the enhancement factor, $\tau_{\mathrm{e}}$ is the effective deviatoric stress, $n$ is the creep exponent, and $A$ is the ice softness which depends on temperature, pressure, and water content of the ice. The enhancement factor and the creep exponents are taken to be different for the SIA and the SSA. We use $E_{\mathrm{SSA}} = 1.3, n_{\mathrm{SIA}} = 3$ while varying $E_{\mathrm{SIA}}$ and $n_{\mathrm{SSA}}$ following Aschwanden and Brinkerhoff (2022). The numerical value of A in eqn. 6 is not changed, only the units are adjusted.

The basal sliding velocity $\mathbf{u}_{\mathrm{b}}$ in the SSA is related to the basal shear stress $\tau_{\mathrm{b}}$ through the pseudo-plastic power law:

$$
\tau_{\mathrm{b}} = -\tan(\varphi)N_{\mathrm{till}}\frac{\mathbf{u}_{\mathrm{b}}}{u_{\mathrm{th}}^q|\mathbf{u}_{\mathrm{b}}|^{1-q}}, \tag{7}
$$

where $q$ is the sliding exponent and $u_{\mathrm{th}} = 100$ m a$^{-1}$ is a characteristic speed. The till friction angle, $\varphi$, is parameterized as a continuous function of bedrock topography that increases linearly from $\varphi_{\mathrm{min}}$ to $\varphi_{\mathrm{max}}$ between $z_{\mathrm{min}}$ and $z_{\mathrm{max}}$. The effective pressure on the till, $N_{\mathrm{till}}$, decreases exponentially with the water level in the till, $W_{\mathrm{till}}$ (Bueler and van Pelt, 2015):

$$
N_{\mathrm{till}} = \min\left\{P_0, \tilde{N}_0\left(\frac{\delta P_0}{\tilde{N}_0}\right)^{W_{\mathrm{till}}/W_{\mathrm{till}}^{\mathrm{max}}}\right\}. \tag{8}
$$

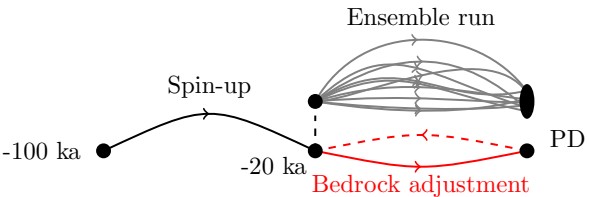

**Figure 2.** Schematic of the model ensemble experiment. The ice sheet is initialized at -100 ka using present-day geometry and run through the last glacial period at 20 km resolution. The bedrock is then iteratively updated at -20 ka to reduce the modeled present-day bedrock topography deviation. After finding a suitable bedrock topography, the ice-sheet model is branched off at -20 ka, and an ensemble of simulations is run at 10 km resolution. While the x-axis depicts time, the y-axis is only used to reflect that the states differ.

It decreases to a fraction $\delta$ of the overburden pressure, $P_0$, when the water level reaches its maximum value, $W_{\text{till}}^{\text{max}} = 2$ m. The parameter $\tilde{N}_0 = 5.6 \times 10^8$ Pa represents the effective pressure the till would have at zero water content if it were not capped
by the overburden pressure.

## 2.5   Earth deformation and initialization

The bedrock responds to changes in ice load by the visco-elastic bed deformation model by Lingle and Clark (1985) and Bueler et al. (2007) with flexural rigidity $D = 5 \times 10^{24}$ N m and upper mantle viscosity $\eta = 1 \times 10^{21}$ Pa s.

We initialize the GrIS by running the model from -100 ka to -20 ka (all times are relative to 2 ka CE) at 20 km grid resolution
using the parameters listed in Table 1 and with initial bedrock topography taken to be the same as the present day (Morlighem, 2022; Jakobsson et al., 2020; GEBCO Bathymetric Compilation Group, 2023).

Since the modeled ice extent, and consequently the surface elevation, is sensitive to ocean melt and sea level forcing, we apply an artificial correction to the bed topography to ensure that the present-day ocean mask closely aligns with observations.

To achieve this, we iteratively adjust the bedrock at -20 ka (as illustrated in Fig. 2) so that the modeled bedrock topography
at the end of the simulation better matches the observed present-day topography. A simulation with 20 km resolution is run from -20 ka to the present, and the deviation between the modeled and observed bedrock topography is used to update the initial bedrock according to

$$b_{i+1}^0 = b_i^0 + K(b^{\text{obs}} - b_i^1), \tag{9}$$

where $b^{\text{obs}}$ is the observed present-day topography (Morlighem, 2022; Jakobsson et al., 2020; GEBCO Bathymetric Compila-
tion Group, 2023), $b_i^0$ is the modeled bedrock topography at -20 ka, and $b_i^1$ is the modeled bedrock topography at the present day. This iterative correction method is similar to the approach used by van Calcar et al. (2023), who employed a comparable scheme to improve present-day topography. The relaxation parameter $K = 0.7$ is introduced to prevent overcompensation from any potential positive feedback associated with the updated bedrock, although such feedback may not be significant, given that

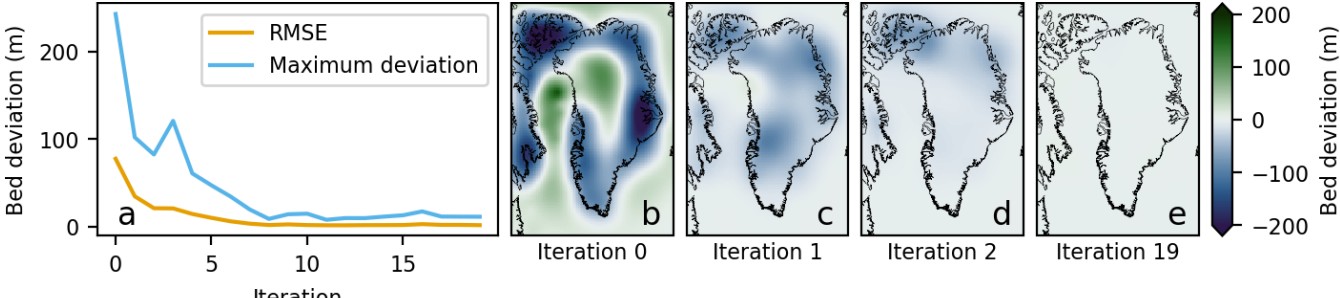

**Figure 3.** Iterative bedrock adjustment. (a) The modeled present-day bedrock elevation deviation compared to Morlighem (2022), Jakobsson et al. (2020), and GEBCO Bathymetric Compilation Group (2023). (b–e) shows the zeroth, first, second, and 19th iteration of the modeled present-day bedrock elevation deviation from observed.

the bedrock–mass balance feedback is likely negative. After 20 iterations, the RMSE of the bedrock decreased from 77.4 m to 3.3 m, as shown in Fig. 3. The impact of these iterations on the surface elevation is illustrated in Fig. A6.

At -20 ka, the simulation is branched using the adjusted bedrock ($b^0$) from the last step (shown in Fig. A8), and an ensemble of simulations is run at a 10 km grid resolution until the present day, varying the 20 parameters listed in Table 1.

## 3  Bayesian inference

To account for model uncertainty and assess the importance of model parameters, we run an ensemble of simulations from -20 ka to the present, varying the 20 parameters listed in Table 1. The dynamic parameters are based on those varied by Aschwanden and Brinkerhoff (2022), while the oceanic and atmospheric parameters are introduced in sections 2.2 and 2.3.

To effectively sample the parameter space, 841 parameters are drawn using the second-order orthogonal Latin Hypercube Sampling (LHS) design (Tang, 1993). This ensures that all pairs of parameters are sampled uniformly and reduces the risk of clustering.

For each of the four ice-core sites, we calculate the likelihood of observing the ice-core-derived elevation history given the modeled elevation change with model parameters $\mathbf{m}$:

$$\rho(\mathbf{h}_i^{\mathrm{obs}}|\mathbf{m}) \propto \prod_j \exp\left(-\frac{\left(h_{i,j}(\mathbf{m}) - h_{i,j}^{\mathrm{obs}}\right)^2}{2\sigma_i^2\beta}\right), \tag{10}$$

where $i$ denotes the ice-core site, and $j$ is the time step of the ice core samples to which modeled elevations are interpolated. $\sigma_i$ are the uncertainties from Vinther et al. (2009), derived from the spread of $\delta^{18}O$ values in two parallel records from the Agassiz Ice Cap and the uncertainty in bedrock uplift at the Agassiz and Renland sites. The lapse rate uncertainty used to derive elevation changes is not included. Present-day and past elevations are weighted equally to avoid biasing the likelihoods toward the present configuration. Following Aschwanden and Brinkerhoff (2022), we introduce $\beta = 100$ to account for autocorrelation

| Parameter | Description | Spin-up | Range | Estimates | | | | |
|---|---|---|---|---|---|---|---|---|
| | | | | Combined | CC | NGRIP | GRIP | Dye 3 |
| **Atmosphere** | | | | | | | | |
| $\Delta T$ | Temperature reconstruction | 3 | 1—5 | 1 | 1 | 4 | 2 | 3 |
| $f_s$ | PDD param. snow (mm $K^{-1}$ $d^{-1}$) | 5.04 | 5.7—8.9 | 6.5±0.7 | 7.1±0.8 | 7±1 | 7.5±0.7 | 7.8±1.0 |
| $f_i$ | PDD param. ice (mm $K^{-1}$ $d^{-1}$) | 12.5 | 7—10 | 7.7±0.6 | 8.5±0.8 | 8.6±0.8 | 8.7±0.9 | 8.4±0.7 |
| $\Gamma$ | Atmospheric lapse rate (K $km^{-1}$) | 5 | 4—9 | 5.4±0.7 | 7±1 | 6±1 | 6±1 | 6±1 |
| $\omega_\downarrow$ | Southern precip. scaling (% $K^{-1}$) | 5 | 0—4.5 | 2±1 | 2±1 | 2±1 | 2±1 | 2±1 |
| $\omega_\uparrow$ | Northern precip. scaling (% $K^{-1}$) | 7 | 0—9 | 2±1 | 3±2 | 2±1 | 4±2 | 3±2 |
| **Ocean** | | | | | | | | |
| $H_{cr}$ | Threshold for thickness calving (m) | 50 | 50—150 | 96±17 | 87±25 | 97±30 | 112±30 | 106±30 |
| $\sigma_{max}$ | Characteristic stress (MPa) | 1 | 0.8—1.2 | 0.92±0.09 | 1.03±0.10 | 1.0±0.1 | 1.0±0.1 | 1.0±0.1 |
| $\dot{m}^o_\downarrow$ | Melt rate south of 71°N (m $a^{-1}$) | | 300—500 | 394±29 | 409±65 | 400±56 | 391±56 | 391±50 |
| $\dot{m}^o_\uparrow$ | Melt rate north of 80°N (m $a^{-1}$) | | 10—30 | 20±6 | 19±6 | 21±6 | 19±6 | 19±5 |
| $\tau$ | Ocean melt onset (ka) | | 4—8 | 5.6±0.6 | 5.5±0.7 | 6±1 | 6±1 | 6±1 |
| $\Delta\tau$ | Ocean melt set in time (ka) | | 0—2 | 1.1±0.5 | 1.1±0.6 | 0.9±0.6 | 1.0±0.6 | 0.7±0.6 |
| **Dynamics** | | | | | | | | |
| $n_{SSA}$ | Creep exponent for the SSA (1) | 3.3 | 3.2—3.4 | 3.28±0.04 | 3.35±0.04 | 3.33±0.04 | 3.25±0.04 | 3.23±0.03 |
| $E_{SIA}$ | Enhancement factor for the SIA (1) | 3 | 2.5—3.3 | 3.0±0.2 | 2.7±0.2 | 2.9±0.2 | 3.1±0.2 | 3.1±0.2 |
| $q$ | Basal sliding power coefficient (1) | 0.8 | 0.7—0.9 | 0.82±0.05 | 0.79±0.05 | 0.81±0.06 | 0.80±0.06 | 0.83±0.06 |
| $\delta$ | Effective pressure parameter (%) | 2 | 1.5—2.5 | 2.1±0.2 | 2.0±0.2 | 2.0±0.3 | 2.0±0.3 | 2.1±0.3 |
| $\varphi_{min}$ | Minimal till friction angle (°) | 10 | 5—10 | 8±1 | 8±1 | 7±1 | 7±1 | 8±1 |
| $\varphi_{max}$ | Maximal till friction angle (°) | 42 | 40—45 | 43±1 | 43±2 | 42±1 | 43±1 | 43±1 |
| $z_{min}$ | Till friction cutoff elevation (m) | -700 | -600—-300 | -421±60 | -481±80 | -443±84 | -433±80 | -447±77 |
| $z_{max}$ | Till friction cutoff elevation (m) | 700 | 0—500 | 271±172 | 238±118 | 231±134 | 252±154 | 271±146 |

**Table 1.** List of the 20 parameters that are varied in our ensemble of simulations. The temperature reconstruction is sampled discretely. The estimated parameter values are given as the mean plus minus the standard deviation of the posterior PDFs except for the temperature reconstruction, which is the mode of the posterior PDFs.

in the uncertainties, reducing the degrees of freedom by a factor of 100, which corresponds to a decorrelation time of 2000 years.

Additionally, we calculate the combined likelihood of the ice-core-derived elevation changes for all sites, which is proportional to the product of the site-specific likelihoods, assuming no spatial correlation between the drill sites:

$$\rho(\mathbf{h}^{obs}|\mathbf{m}) \propto \prod_{i=1}^{4} \rho(\mathbf{h}_i^{obs}|\mathbf{m}). \tag{11}$$

The parameters are sampled uniformly over the ranges specified in Table 1, which focuses on the volume of parameter space that shows the highest likelihood in an initial ensemble of simulations.

The posterior joint probability density functions (PDFs) are then given by Bayes's theorem:

$$\rho(\mathbf{m}|\mathbf{h}_i^{\text{obs}}) = \frac{\rho(\mathbf{h}_i^{\text{obs}}|\mathbf{m})}{\rho(\mathbf{h}_i^{\text{obs}})}\rho(\mathbf{m}), \tag{12}$$

where the prior distribution $\rho(\mathbf{m})$ is taken to be uniform within the intervals listed in Table 1. From the five posteriors, we get five PDFs of ice sheet evolution through the Holocene from which we estimate relevant observables listed in Table 2. Unless stated otherwise, all model results are based on the combined posterior PDF.

To evaluate the effectiveness of the sampling, we compute the effective sampling size for each of our five normalized posteriors:

$$n_{\text{eff}} = \frac{1}{\sum_k \rho(\mathbf{m}_k|\mathbf{h}_i^{\text{obs}})^2}, \tag{13}$$

where $k$ denotes the sample member. The effective sample size is the number of equally weighted samples that would yield the same variance of the mean as the weighted set of samples. If only one ensemble member has a non-zero likelihood, the effective sample size is one. Conversely, if all members have the same likelihood, the effective sample size equals the actual sample size, namely 841.

Although constraining the simulations to present-day observations would increase confidence in our modeled present-day state, we avoid doing so because this study focuses on the transient evolution of the GrIS and how the ice-core-derived surface elevation histories from Vinther et al. (2009) can be used to constrain the ice sheet's evolution.

## 4 Results

### 4.1 Surface elevation evolution

Figure 4 shows the modeled and ice-core-derived surface elevations during the Holocene at the four ice-core sites. The site-specific modeled elevations closely match the ice-core-derived reconstructions, reproducing the substantial thinning observed at CC and Dye 3, as well as the more moderate thinning at the interior sites GRIP and NGRIP, with RMSEs ranging from 12 to 53.6 m. The combined elevation estimate is lower than the site-specific estimates at CC and Dye 3 at the onset of the Holocene, while it is too high at GRIP. The corresponding histories of bedrock elevation and ice thickness associated with these surface changes are shown in Fig. A7.

To illustrate the effect of allowing the ice sheet to advance beyond its present-day boundaries, we ran two additional simulations: one restricted from advancing beyond the present-day GrIS coast and another restricted from advancing beyond the ECS mask. Both simulations started from the unrestricted branch-off point at -20 ka and used the same parameters as the ensemble member with the highest combined likelihood, although these parameters may not necessarily be optimal for either of the restricted runs. The simulation restricted to the present-day GrIS coast could not reproduce the observed surface elevation history

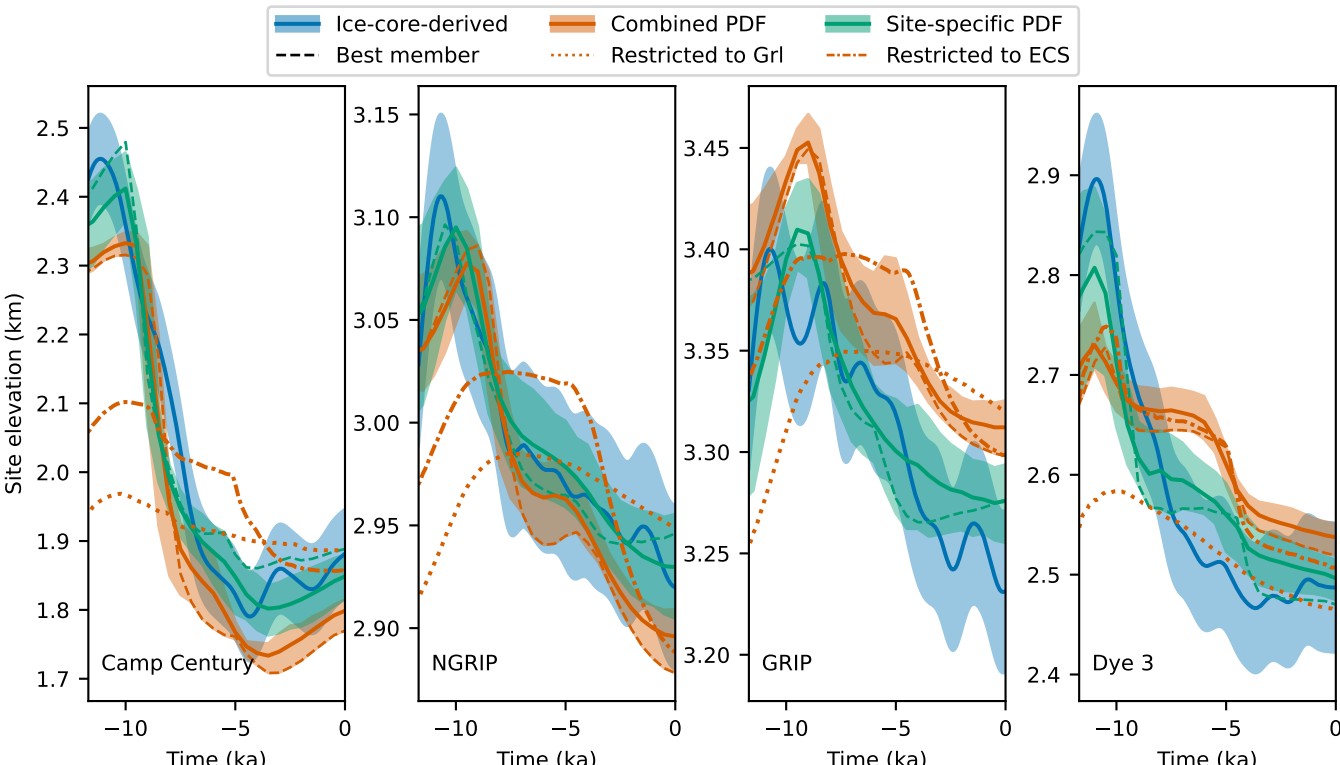

**Figure 4.** Observed and modeled surface elevation over the last 11.7 ka for the ice-core sites Camp Century (CC), NGRIP, GRIP, and Dye 3. The blue lines are the ice-core-derived surface elevations from Vinther et al. (2009), and the blue envelopes denote one standard deviation. The orange solid lines show the combined PDF means, while the green solid lines indicate the site-specific PDF means for the site shown in each panel. Shaded orange and green envelopes represent the corresponding 16th–84th percentile ranges. The dashed lines are the ensemble members with the highest likelihood for each site (green) and the highest combined likelihood (orange). The orange dotted and dash-dotted lines are simulations with the same parameters as the best ensemble member but restricted to ECS (dash-dotted) and the present-day land margin of the GrIS (dotted).

at CC, NGRIP, and Dye 3, showing the importance of a dynamic grounding line. The simulation restricted to not advancing beyond the ECS performed better but also failed to reproduce the thinning at CC, showing the effect of including the Canadian Arctic Archipelago when modeling the GrIS Holocene history. The RMSEs associated with the four ice-core-derived elevation histories are listed in Table 2 for the five estimates.

## 4.2 Inferred parameters

The ice-core-derived surface elevation histories provide constraints on the model parameters, with the marginal PDFs for each of the five posteriors shown in Fig. 5. The degree to which individual parameters are constrained varies, and the estimated values are summarized in Table 1.

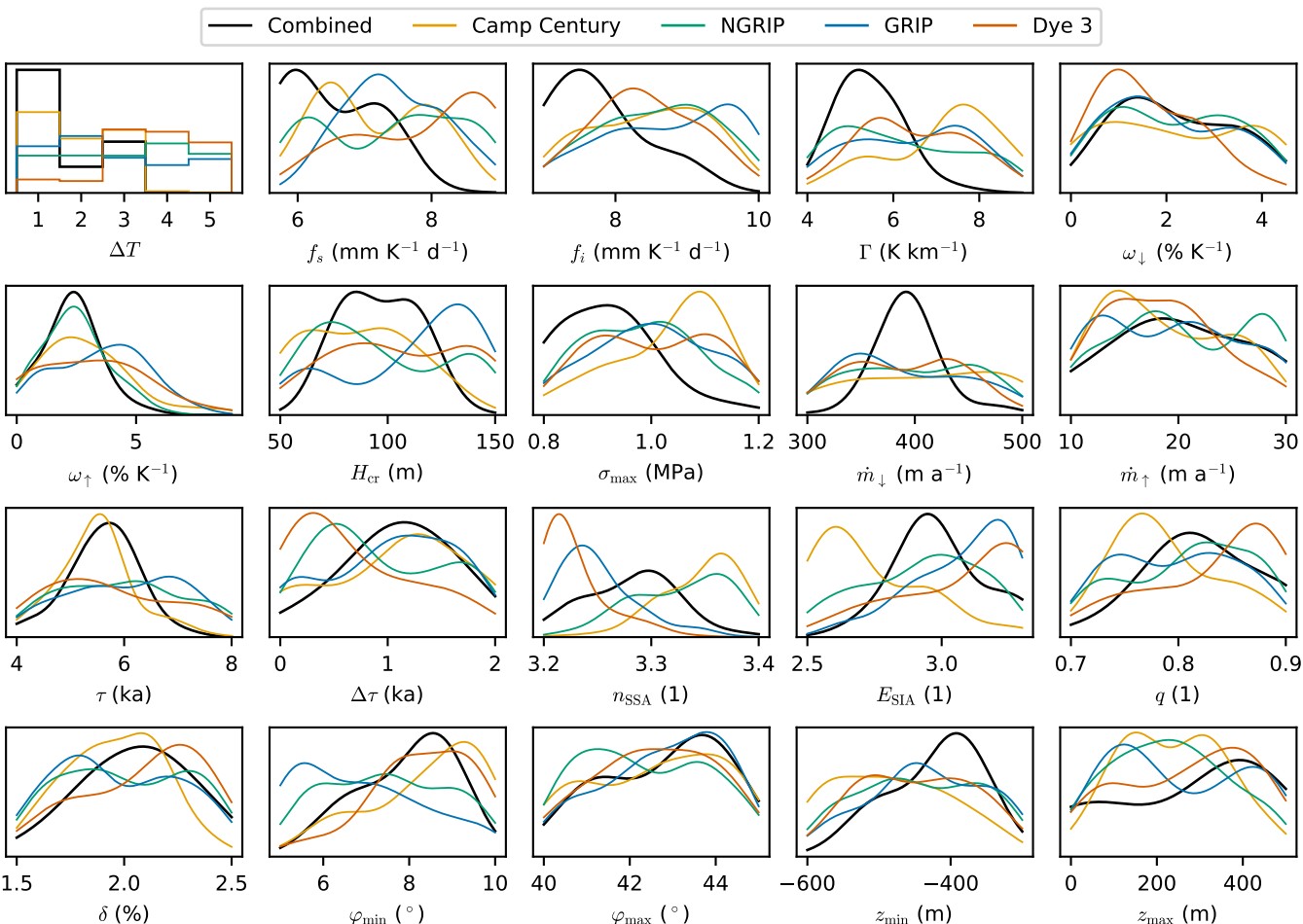

**Figure 5.** Kernel density estimates of the inferred marginal PDFs for the 20 model parameters that we varied. The units on the y-axes are the inverse of that of the x-axes.

Notably, the site-specific and combined estimates of the SIA enhancement factor, $E_{SIA}$, differ substantially. The site-specific estimate based on CC is $2.7\pm0.2$, while that based on GRIP is $3.1\pm0.2$. Similarly, the site-specific estimates of the SSA creep exponent, $n_{SSA}$, differ, with higher values for CC and NGRIP than for GRIP and Dye 3.

Among the five temperature reconstructions, reconstruction 1, being the coldest throughout the Holocene, has the highest combined probability (61%) and the highest site-specific probability for CC (40%), where the largest surface thinning is observed. In contrast, reconstruction 2 is more than one degree warmer during the early Holocene and performs worse than reconstruction 1. Reconstructions 4 and 5, the warmest of the set, have near-zero probability at CC but perform better at other sites; notably, reconstruction 4 has the highest site-specific probability for both NGRIP and Dye 3.

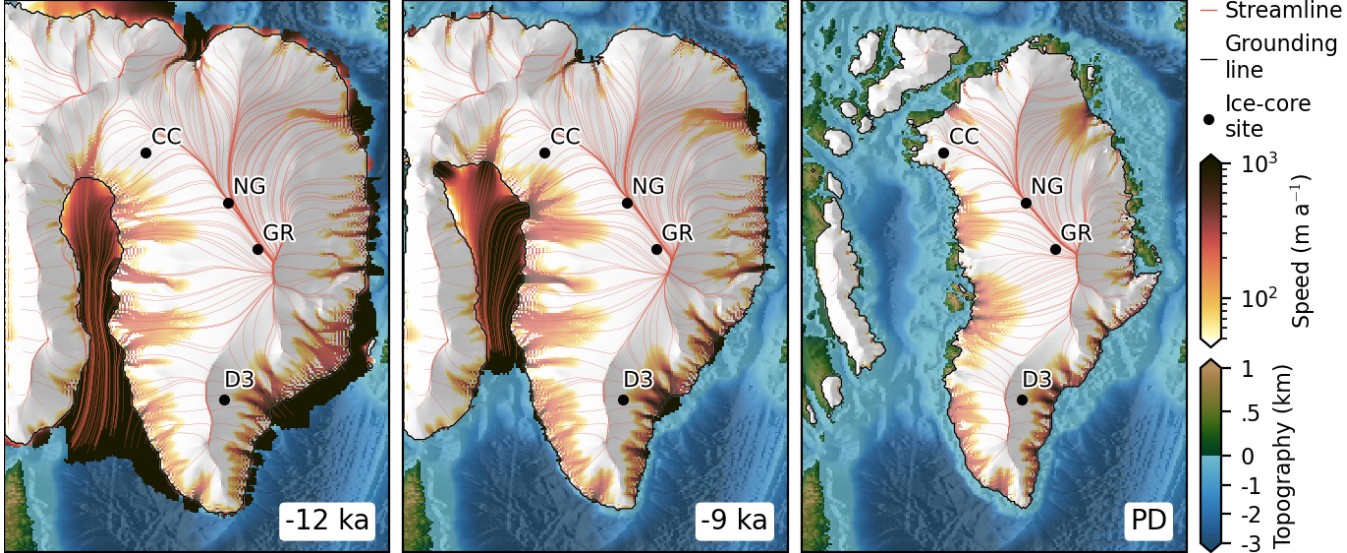

**Figure 6.** Time slices showing modeled surface speed, streamlines, bed topography, and ice shelf extent for the ensemble member with the highest combined likelihood at -12 ka, -9 ka, and present day (PD). The present-day locations of the ice-core sites Camp Century (CC), NGRIP (NG), GRIP (GR), and Dye 3 (D3) are shown at the top.

The northern precipitation parameter, $\omega_\uparrow$, is more constrained by the northern sites CC and NGRIP, where it has the most
influence. Likewise, the southern precipitation parameter, $\omega_\downarrow$, is most constrained by Dye 3. Both parameters are estimated to be $2\pm1\%$ $K^{-1}$ which is substantially lower than the default of $7.3\%$ $K^{-1}$ introduced by Huybrechts (2002) resulting in less accumulation in the warm periods of the Holocene and more accumulation in the cold glacial where the ice sheet builds up.

The onset of sub-shelf ocean melt is well constrained by CC, occurring at $5.6\pm0.6$ ka before present.

### 4.3 Modeled Holocene evolution

From the branch-off point at -20 ka until the onset of the Holocene (11.7 ka ago), the modeled ice sheet bridges the gap between Canada and Greenland across the Baffin Bay and the Nares Strait. Figure 6 shows the ice sheet configuration at -12 ka, -9 ka, and the present day, while Fig. 7 presents the volume and area evolution from the branch-off point to the present. The model clearly responds to the change in resolution at the branch-off point, showing a positive drift in volume, though this shock appears to have stabilized before the start of the Holocene. By -12 ka, the ice sheet reaches its glacial maximum extent,
grounding on the continental shelf and through the Nares Strait.

At -12 ka, the GrIS has a modeled grounded area of $2.96\pm0.03 \times 10^6$ km², within the ECS. This is 49% or $0.98\pm0.05 \times 10^6$ km² larger than the present-day modeled area and it is 0.9% larger than the minimum LGM extent and 5.6% smaller than the maximum LGM extent from Leger et al. (2024). Compared to the modeled present-day GrIS, the modeled grounded volume is

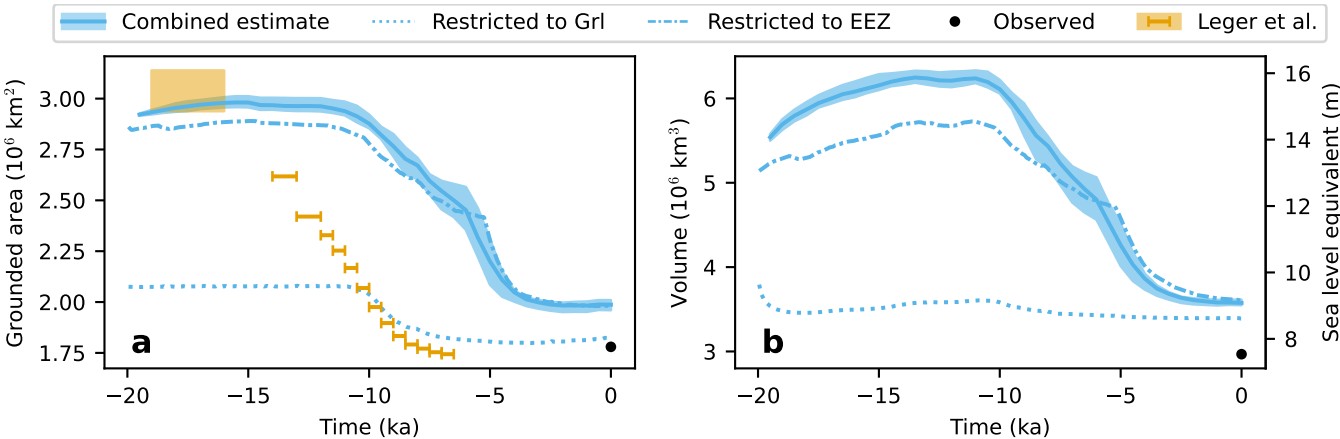

**Figure 7.** Modeled evolution of the GrIS grounded area (a) and volume (b), including peripheral glaciers. The blue shaded area denotes the estimated standard deviation. Present-day values from Bamber et al. (2013) are shown as a black dot. The estimated area excluding peripheral glaciers from Leger et al. (2024) is shown as a square for the LGM extent and with error bars for the dated isochrones, for comparison.

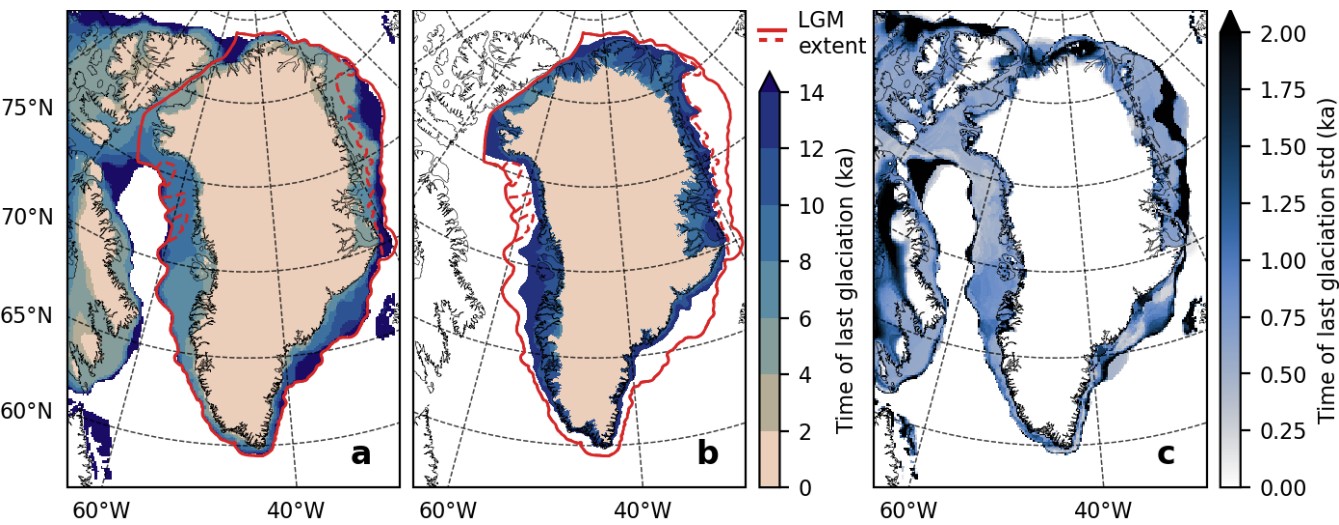

**Figure 8.** Isochrones showing the time of last glaciation for the model (a) and for the empirical reconstruction of Leger et al. (2024) (b). The red lines mark the maximum (solid) and minimum (dashed) LGM extent from Leger et al. (2024). (c) shows the modeled standard deviation of the time of last glaciation.

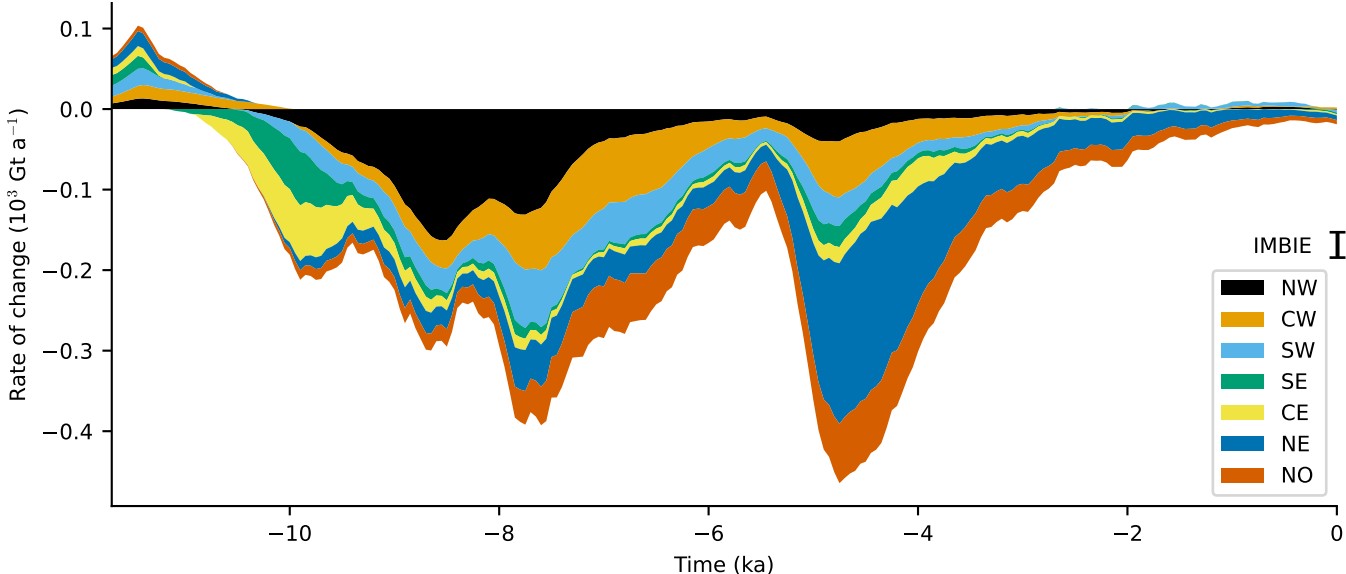

**Figure 9.** Rate of change of grounded ice by basin for the ensemble member with the highest likelihood. The mass change is smoothed using a running mean of 500 years, then divided into gain and loss, and then accumulated by basin. The 1992-2020 estimated mass loss rate from The IMBIE Team (2020) is shown for comparison.

6.6±0.4 m SLE larger at -12 ka. Additionally, the grounded volume above flotation is 5.3±0.3 m SLE larger, which contributed
to the global mean sea-level rise. The modeled times of last glaciation is shown in Fig. 8.

Outside the ECS the IIS and Laurentide Ice Sheet are cut off at the domain boundary with a Dirichlet boundary condition of zero thickness. This moves the ice divide at Baffin Island further to the east than if it had been connected to a complete Laurentide Ice Sheet. Together they have a grounded area of $1.20\pm0.03 \times 10^6$ km$^2$ and a grounded volume of 5.0±0.2 m SLE.

During the Holocene collapse of the IIS, the ice divide at the GrIS moves towards the west and the ice streams reorganize
in northern Greenland as shown in Fig. 6. This divide migration could explain the onset of NEGIS, as found by Franke et al. (2022), and the shutdown of the older, more northern ice stream, as observed by Jansen et al. (2024).

Figure 9 shows the rate of change of grounded ice for the ensemble member with the highest combined likelihood for the seven basins of the GrIS. The GrIS rate of change becomes negative at -10.7 ka and exhibits two distinct peaks: one at -7.8 ka, with a mass loss rate of 548 Gt a$^{-1}$, and another at -4.95 ka, following the onset of sub-shelf melting, with a mass loss rate
of 511 Gt a$^{-1}$. It continues to be negative for the rest of the Holocene except for a few times during the last 2 ka, where the average mass loss rate is 23.7 Gt a$^{-1}$. The mass loss rates stated here are averaged over 50 years.

### 4.4 Present-day configuration

The modeled present-day extent of grounded ice deviates from the observed extent, as shown in Fig. 10a. Most notably, the modeled extent is larger in the Canadian Archipelago, while it fails to cover an area of $0.08\pm0.01 \times 10^6$ km$^2$ and inaccurately

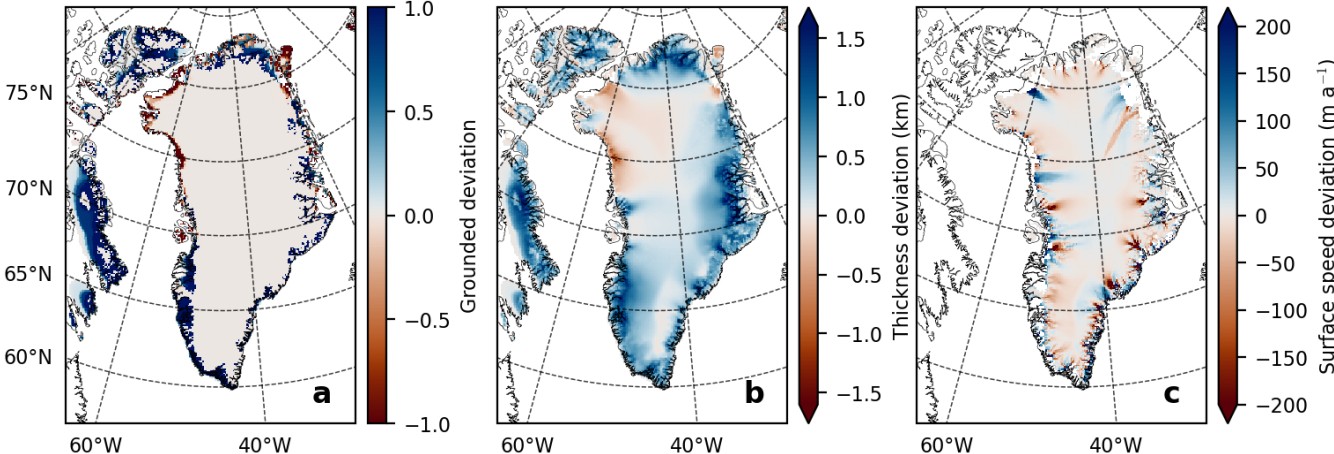

**Figure 10.** Difference between modeled and observed ice sheet (modeled - observed). (a) Difference in present-day grounded ice extent, where a value of one indicates grounded ice and zero indicates ice-free areas (Morlighem, 2022; RGI Consortium, 2023). (b) Difference in ice thickness (Morlighem, 2022; Millan et al., 2022). (c) Difference in surface speeds (Solgaard et al., 2021).

covers an area of $0.19\pm0.01 \times10^6$ km$^2$ compared to the observed GrIS extent. This comparison includes peripheral glaciers and excludes ice thinner than 10 meters, which is considered to be seasonal.

The modeled GrIS grounded volume at present day is $9.1\pm0.1$ m SLE, which is 1.5 m SLE larger than the observed grounded volume, including peripheral glaciers (Morlighem, 2022). This discrepancy can be attributed to the ice thickness deviations at the GrIS margin, as shown in Fig. 10b.

In the northwest, the modeled ice sheet is thinner than observed at the Humboldt Glacier, resulting in an overestimation of surface speed. In the northeast, the model fails to reproduce the flow pattern of NEGIS and instead simulates a faster-flowing ice stream located farther north. These discrepancies are illustrated by the deviations in modeled surface speeds shown in Fig. 10c, compared to observations by Solgaard et al. (2021).

The modeled present-day uplift rates deviate from the GPS-derived GIA uplift rates reported by Schumacher et al. (2018),
as shown in Fig. 11a. The largest deviations occur in the area formerly covered by the IIS, where the modeled present-day bedrock topography differs by up to 93 m from observations (Fig. 11b), with a root mean square error (RMSE) of 27 m. The total modeled uplift from -12 ka to the present reaches a maximum of 509 m in the IIS region (Fig. 11c). At Agassiz and Renland, the modeled bedrock uplifts are $345\pm9$ m and $168\pm9$ m, respectively. These values are slightly higher than the 275 m and 110 m uplifts, respectively, used by Vinther et al. (2009) in their surface elevation reconstructions.

| Observable | Estimates | | | | | | Restricted | |
|---|---|---|---|---|---|---|---|---|
| | Combined | CC | NGRIP | GRIP | Dye 3 | Prior | Grl | ECS |
| **Elevation history RMSE** | | | | | | | | |
| CC (m) | 88.1 | 53.6 | 116.8 | 154.3 | 170.7 | 119 | 223.4 | 165.9 |
| NGRIP (m) | 26.8 | 44.9 | 12 | 69.8 | 69.4 | 34.9 | 67.8 | 49.9 |
| GRIP (m) | 58.6 | 123 | 86 | 27.2 | 29.5 | 62 | 64.5 | 59.5 |
| Dye 3 (m) | 99.1 | 153.4 | 116.4 | 87.2 | 55.4 | 106.1 | 123.5 | 88.5 |
| **Present-day configuration** | | | | | | | | |
| Grounded ice volume (m SLE) | 9.0±0.1 | 9.5±0.3 | 9.1±0.3 | 8.7±0.3 | 8.7±0.2 | 9.0±0.4 | 8.55 | 9.08 |
| Grounded area ($10^6$ km$^2$) | 1.99±0.02 | 2.00±0.03 | 1.94±0.05 | 1.95±0.05 | 1.93±0.04 | 1.95±0.06 | 1.82 | 1.98 |
| Falsely grounded ($10^6$ km$^2$) | 0.19±0.01 | 0.21±0.02 | 0.17±0.03 | 0.18±0.03 | 0.17±0.03 | 0.18±0.04 | 0.07 | 0.19 |
| Missing grounded ($10^6$ km$^2$) | 0.08±0.01 | 0.08±0.01 | 0.11±0.02 | 0.11±0.02 | 0.12±0.02 | 0.10±0.02 | 0.13 | 0.08 |
| Ice thickness RMSE[†] (m) | 420.8 | 495.2 | 435.9 | 396.4 | 394.9 | 428 | 313.4 | 418.7 |
| Bed topography RMSE (m) | 27 | 42.2 | 19 | 18.5 | 17.5 | 20.2 | 58.3 | 58.3 |
| Surface speed RMSE[†] (ma$^{-1}$) | 84.4 | 80.7 | 79.5 | 82.7 | 82 | 79.1 | 95.3 | 96.3 |
| **Configuration at -12 ka** | | | | | | | | |
| Grounded ice volume (m SLE) | 15.7±0.3 | 16.0±0.7 | 15.8±0.5 | 15.1±0.6 | 15.8±0.7 | 15.3±0.9 | 9.03 | 14.31 |
| Grounded area ($10^6$ km$^2$) | 2.96±0.03 | 2.93±0.03 | 2.93±0.04 | 2.96±0.05 | 3.01±0.05 | 2.93±0.05 | 2.08 | 2.87 |
| dvdt last 500 a (mm SLE ka$^{-1}$) | -23±26 | -31±27 | -74±133 | -52±106 | -75±143 | -60±112 | -18.19 | -70.81 |
| Time of collapse (ka b2k) | 4.9±0.5 | 4.9±0.7 | 6±1 | 6±1 | 5±1 | 6±1 | | |
| $n_{\text{eff}}$ | 9.28 | 41.36 | 151.87 | 99.67 | 28.56 | 841.00 | | |

**Table 2.** Estimates of key observables for the past and present of the GrIS, as well as observables for the simulations restricted to the present-day Greenland mask (Grl) and the ECS. All observables are calculated within the ECS mask at model resolution and do not include Canada.
[†] RMSEs are calculated within the present-day observed grounded mask.

# 5 Discussion

## 5.1 Collapse of the Innuitian ice bridge and interior thinning

The focus of this study is to reproduce surface elevation histories at the ice core locations in interior Greenland. To accurately model the early Holocene thinning in northwest Greenland, we included the Canadian Arctic Archipelago in our domain and allowed the ice sheet to advance beyond its present-day boundaries. With these model choices, the ice bridge across the Nares Strait that connected the Greenland Ice Sheet (GrIS) and the Innuitian Ice Sheet (IIS) during the Last Glacial Maximum forms in the model, as well as the ice shelf that covered Baffin Bay. This, in turn, enables the ice sheet to grow thicker in Greenland's interior during the Last Glacial Maximum at the four ice-core sites, due to the development of the ice bridge. This is demonstrated by comparing two simulations: one constrained to the present-day land margin and the other to the ECS (Fig. 4).

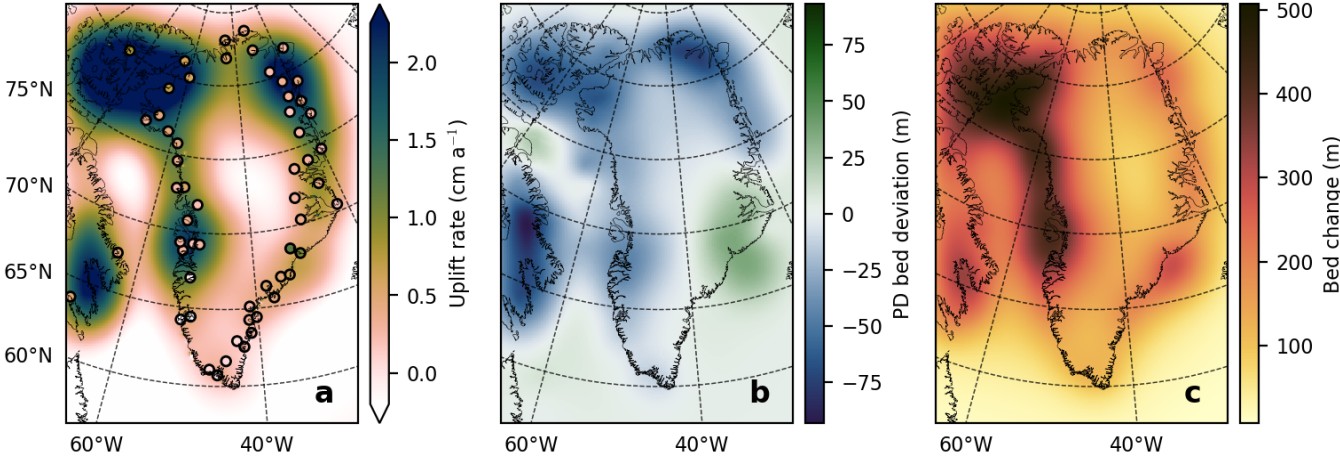

**Figure 11.** (a) Modeled present-day bedrock topography uplift rates and GPS-derived uplift rates from Schumacher et al. (2018). (b) Modeled present-day bedrock topography deviation from observed. (c) Modeled bed uplift between -12 ka and present-day.

In our model, the IIS is connected to the GrIS across Nares Strait during the glacial maximum. From the saddle point between the ice sheets, the ice flow diverges into two oppositely flowing ice streams in the Nares Strait: one flowing southwestward, which is similar to the Smith Sound Ice Stream suggested by England et al. (2006), and another flowing northeastward. The southwestward ice stream discharges into Baffin Bay, which is covered by an extensive ice shelf, as proposed by Couette et al. (2022), that provides buttressing for the western GrIS.

In our simulations, deglaciation occurs in two stages (Fig. 9). The ice sheet begins to thin and retreat in response to the abrupt temperature rise at the transition at -11.7 ka. Increased surface melting drives the first stage of mass loss in the early Holocene, prior to the onset of ocean forcing. This enhanced melting leads to the collapse of the ice shelf in Baffin Bay around -9 ka, reducing the buttressing effect and causing further thinning and retreat of the ice sheet. Just before -9 ka, a large paleo–ice stream develops inland from Baffin Bay, accelerating ice flow along the northwest coast into the bay and driving additional inland thinning in this region. Mass loss slows after -8 ka, before ocean forcing begins at -5.6 ka, initiating the second stage of accelerated mass loss (Fig. 9). This culminates in the collapse of the ice bridge across Nares Strait at -4.9±0.5 ka.

The timing of the modeled collapse of the ice bridge in Nares Strait occurs more recently than suggested by geological evidence. In our simulation, the collapse takes place at -4.9±0.5 ka, approximately 0.7 ka after the onset of sub-shelf melting and about 3 ka later than the estimate by England et al. (2006). Additionally, the lateral retreat of the ice margin also occurs several thousand years later than found by Leger et al. (2024), as shown in Figs. 7 and 8.

Our simulation is, however, in good agreement with the surface elevation lowering at the CC ice-core site in northern Greenland, where the Holocene thinning rate peaks around -9 ka (Fig. 4). The surface lowering at the CC site at -9 ka coincides with the emergence and acceleration of the paleo–ice stream along the Baffin Bay coast (Fig. 6 and Supplementary Material), and occurs several thousand years prior to the collapse of the ice bridge.

At present, many marine-terminating ice streams in the northwest sector of the Greenland Ice Sheet flow into Baffin Bay, bounded by high bedrock topography near Upernavik to the south and Pituffik to the north. Our simulation indicates that paleo–ice streams formed in this sector during the glacial period (Fig. 6a). Around -9 ka, these evolved into a transient, 50 km-wide ice stream terminating in Melville Bay, with a fast-flowing central trunk reaching velocities of several kilometers per year, comparable in size to the present-day NEGIS outlets. This ice stream developed over a bedrock valley system that links the northwestern interior to Melville Bay, rapidly draining the northwest sector of the GrIS and driving a retreat of the ice margin to near its present-day position within approximately 1 ka.

Thus, the primary cause of the surface elevation lowering at the CC ice-core site is the formation of the paleo–ice stream inland from Baffin Bay, rather than the collapse of the ice bridge. This is a novel finding of our study, which links to the work by Tabone et al. (2024), who associated Holocene elevation changes in interior Greenland with an acceleration of the paleo NEGIS.

Further work, beyond the scope of this study, may be able to reconcile the modeled ice sheet evolution with both the ice-core-derived interior surface elevation histories and the lateral retreat and timing of the ice bridge collapse inferred from geological evidence. It should be noted, however, that ensemble members with earlier onsets of sub-shelf ocean melting, while leading to earlier retreat, result in the ice sheet becoming too thin at the CC site during the mid-Holocene, whereas those with later onsets remain too thick at present (Fig. A5). This issue might be resolved if the precipitation in early Holocene was higher than assumed here.

## 5.2 Holocene evolution and ice mass loss

Overall, our simulations show that the deglaciation in Greenland occurred between around 10 ka and 3.5 ka, where the total area and volume of the GrIS dramatically decreased from its glacial maximum values to approximately the present-day volume and extent (Fig. 7). The minimum ice-covered area occurred approximately at -2 ka and slightly increased towards the present, while the ice volume has remained relatively constant over the last millennia. Our simulation shows no clear evidence of a minimum ice volume during the Holocene Thermal Maximum. It ends at the present day with a simulated area and volume that exceed the observed values by 5.9% and 20.5%, respectively (Fig. 7).

A previous study by Nielsen et al. (2018) showed that the evolution of the GrIS depends on the assumed climate history through the Holocene. Nielsen et al. (2018) found that the GrIS retreated to a smaller than present-day volume at around 8 ka ago when forced by temperature anomalies that contain the Holocene Thermal Maximum, but their simulations did not include Canada in the domain, and thus initiated their simulation with a GrIS of similar size as at present day. In our simulations, we used the same climate forcing histories as in the study by Nielsen et al. (2018), but we do not find a similar minimum in our simulations for the ensemble members that include the Holocene Thermal Maximum, most likely because the GrIS is too far from equilibrium during the Holocene Thermal Maximum due to the large initial ice sheet. In fact, the simulations that best fit all surface elevation histories are those forced with the climate reconstruction history number 1 (see Fig. 5), which did not show any Holocene Thermal Maximum. For this climate reconstruction, our simulated Holocene ice volume follows a similar pattern as found in Nielsen et al. (2018).

The spatial pattern of mass loss rates from the GrIS has shifted significantly during the Holocene (Fig. 9). In the earliest part of the Holocene, the rate of mass change was slightly positive in all basins due to an increase in snow accumulation over the GrIS. During the first deglaciation phase between 10 and 5.5 ka ago, the mass loss rate was large in all basins, with the largest mass loss rate in the northwest basin, being about the same rate as all other basins combined. The central west basin also had significant mass loss rates, followed by the north and southwest basins. Towards 5.5 ka, the mass loss rates decreased towards zero, which is also seen in the volume record as a temporary stabilization. Between 5.5 ka and 3.5 ka, after the onset of the sub-shelf melt, a second phase in the deglaciation occurred, with a total higher mass loss rate than the first phase, and now dominated by high mass loss rates from the northeast basin and to a lesser extent from the north and central west basins. These two deglaciation phases are also seen in the total volume (Fig. 7b), with a kink around 5.5 ka ago separating the two phases.

Our simulated Holocene mass loss rates exceed the mass loss rates estimated in a previous study (Briner et al., 2020). Briner et al. (2020) simulated the Holocene evolution of the CW and SW basins and assumed that regions are representative of the entire GrIS. They found the maximal values of mass loss during the Holocene to be 60 Gt $a^{-1}$ and that it would most likely be exceeded within this century with rates of mass loss of 8.8 to 359 Gt $a^{-1}$ depending on the climate scenario which is less than our maximal Holocene rate of mass loss at 548 Gt $a^{-1}$ for the ensemble member with the highest likelihood. Our results show that the spatial pattern of retreat has shifted geographically during the Holocene, and the mass loss rates from the GrIS basins have peaked thousands of years earlier in the northwest and west than in the northeast. We conclude that one basin cannot be representative of the entire GrIS, and thus, our results are not directly comparable to the results by Briner et al. (2020).

The importance of calibrating the GrIS evolution with paleo constraints is underscored when examining the mass change rates of the GrIS over the last 500 years. These rates range from a decrease of 487 mm $ka^{-1}$ to an increase of 105 mm $ka^{-1}$ across the ensemble of simulations. Excluding the simulations that utilize the temperature reconstruction from Gkinis et al. (2014), where the temperature anomaly peaks at 4.5 K during this period, we find that the remaining mass loss rates primarily depend on the timing of the onset of ocean forcing, $\tau$ (Fig. A4). This relationship exhibits a strong Pearson correlation coefficient of 0.8. Consequently, the estimated mass loss rate shifts from a prior of -12$\pm$40 mm $ka^{-1}$ to a posterior of -23$\pm$26 mm $ka^{-1}$. This adjustment highlights the critical role of paleo calibration in accurately modeling ice sheet dynamics.

## 5.3 Climatic forcing

The atmospheric conditions and spatial patterns of temperature and precipitation during the glacial period and early Holocene were likely quite different from those of the present day. The Laurentide Ice Sheet is thought to have both shielded Ellesmere Island from precipitation and deflected the jet stream, directing more moisture north of the Laurentide from the North Pacific Ocean into the polar regions. This makes the amount of precipitation over Greenland during that time uncertain (England et al., 2006).

Furthermore, the presence of an ice shelf in Baffin Bay likely influenced temperature and precipitation in a spatially non-uniform manner. Additionally, changes in insolation, which were not accounted for here, would have unevenly increased melt, particularly at higher latitudes (e.g. Robinson and Goelzer, 2014).

Non-uniform temperature anomaly products do exist, such as the TraCE-21K climate simulations (Liu et al., 2009) and the derived product of Badgeley et al. (2020), which was assimilated to match ice-core-derived temperatures. However, we chose not to use these products because they were simulated using the surface topography from ICE-5G (Peltier, 2004). Avoiding these inputs ensures that our ice-sheet reconstruction remains independent of previous reconstructions and avoids circular reasoning.

In our setup, we apply uniform temperature anomalies to be consistent with the assumption of Vinther et al. (2009), namely that local temperature offsets result from a Greenland-wide anomaly combined with local elevation feedback. However, by varying the applied temperature anomalies, lapse rate, and the northern and southern precipitation scaling parameters, we explore a range of plausible spatial and temporal variability in SMB. Shortcomings of this model assumption would then likely manifest as differences in the site-specific inferred PDFs of the atmospheric parameters.

Our model setup also adopts a simplified approach to ocean forcing, reflecting both the limited understanding of its temporal evolution and the desire to maintain interpretability. Once the issue of excessive thinning at CC during the mid-Holocene — caused by earlier ocean-forcing onsets — is addressed, it may become feasible to further constrain the model using the lateral retreat reconstruction of Leger et al. (2024). This could be done by introducing regionally distinct onset timings for ocean forcing in western and eastern Greenland, potentially yielding deeper insights into its spatial and temporal variability.

## 5.4 Inferred parameters

The modeled evolution of the ice sheet during the Holocene is influenced by multiple model parameters, which may compensate for each other due to the complexity of the dynamic processes and climate forcings involved. As a result, interactions between these parameters can obscure their individual effects, making it difficult to draw definitive conclusions from the inferred parameter PDFs.

Nonetheless, differences in the site-specific PDFs suggest spatial variations in SMB that our model does not capture. For instance, the surface elevation history at Dye 3 in the south favors reconstructions with a warm Holocene Thermal Maximum, while CC in the north favors a colder Holocene climate. This could indicate regional temperature differences that challenge the assumption by Vinther et al. (2009) that local temperature offsets were solely due to a uniform, Greenland-wide anomaly and elevation feedback. However, these differences might also reflect compensations for other spatial factors, such as variations in ice rheology or basal friction.

The differences in the estimated ice flow parameters may also stem from poorly resolved outlet glaciers, which lead to an underestimation of ice flux. This is often compensated for by increasing the enhancement factors. Notably, both Kangerlussuaq Gletscher in eastern Greenland and Sermeq Kujalleq in western Greenland require a resolution finer than 3.6 km to be properly resolved, as suggested by Aschwanden et al. (2016), which is not feasible for this study. Additionally, it is possible that the ice-core-derived surface elevation records do not provide sufficient constraints for all model parameters.

The orthogonal Latin Hypercube Sampling technique was chosen for its ability to cover the high-dimensional parameter space more uniformly than simple random sampling, thereby reducing estimation errors. The effective sample size of the combined estimate is 9.28 (Table 2), indicating that the weight is concentrated among a few samples, with the two most likely

members contributing 42% to the estimates. To improve this, an adaptive sampling technique could be implemented to increase sampling density in the regions of parameter space that most influence the estimates.

## 5.5 Bedrock uplift

The present-day bedrock RMSE for the ensemble members ranges from 11.9 to 70.9 m. This is an improvement from the
initial RMSE of 77.4 m before any bedrock adjustment, but it is still higher than the RMSE of 1.47 m achieved after the 12th iteration of the bedrock adjustment scheme. The discrepancy may partly arise from the bedrock adjustment being performed at a 20 km resolution, while the ensemble uses a 10 km resolution. The variation in bedrock RMSE suggests that the modeled bedrock topography is highly sensitive to the ice load history. To improve agreement between the modeled and observed present-day bedrock, additional bedrock adjustment iterations could be performed for each ensemble member, though this
would significantly increase computational demands. However, the reported error in Morlighem (2022) is up to 1000 m in the interior, where data coverage is sparse, and the RMSE is 145 m over land. Given these uncertainties, further refining the bedrock topography may provide limited improvements.

    The modeled uplift rates are generally larger than the GPS-derived GIA uplift rates from Schumacher et al. (2018). This discrepancy may be due to the modeled collapse occurring too late, not allowing enough time for the bed to fully relax, or it
could be because the assumed viscosity of the upper mantle is too low. A higher viscosity would result in smaller past elevation changes, leading to an earlier collapse.

    The viscosity of the upper mantle was not varied in our ensemble; instead, we used the default value of $10^{21}$ Pa s from Lingle and Clark (1985). Estimates of viscosity vary across several orders of magnitude (Bagherbandi et al., 2022), with Albrecht et al. (2020) suggesting a plausible range of $10^{20}$ to $10^{22}$ Pa s for Antarctica. Varying the viscosity would further alter the bedrock
topography history of the ensemble members, providing additional justification for adjusting the bedrock for each member individually.

## 5.6 Validity of elevation histories

Our work, and the elevation histories by Vinther et al. (2009), relies on the assumption that changes in moisture sources did not significantly affect the O-18 fractionation across the ice sheet. It does not consider the potential influence of a paleo ice shelf
in Baffin Bay or the presence of the Innuitian Ice Sheet (IIS) on the fractionation process along the moisture transport pathway from ocean source to ice-core site. Ideally, the ice-sheet model should be coupled to an atmospheric model that simulates the transport and isotopic evolution of moisture from ocean evaporation to ice-sheet precipitation. The resulting modeled O-18 values could then be compared directly to the ice-core measurements. Nevertheless, the derived elevation histories used in our study are supported by measurements of total gas content, which provide an independent proxy for pressure — and thus
elevation — at the close-off depth.

    Lecavalier et al. (2013) proposed revised elevation histories, arguing that the bedrock history along the eastern coastline of Ellesmere Island should be used instead of the Agassiz site itself, due to the complicating influence of the Innuitian Ice Sheet (IIS) prior to 8 ka. Building on this, Lecavalier et al. (2017) assumed that the elevation correction to the $\delta^{18}$O signal at

the Agassiz ice cores should be based on coastal sites. This led to revised temperature anomalies for Agassiz, which, in turn, increased the ice-core-derived surface elevation at Camp Century (CC) by 400 m at the onset of the Holocene. This suggests that our model may underestimate the surface elevation at that time, which could potentially be resolved by increasing the SMB during the last glacial period to allow for greater ice sheet thickening. We attempted to use the temperature anomalies from Lecavalier et al. (2017), and while this did result in increased thinning, the modeled present-day elevation became far too low, with the ice sheet retreating excessively far inland in the northwest.

The assumption that the thickness of the Renland ice cap remains constant throughout the Holocene contradicts our model results. We find that the Renland ice cap thins by 399±56 m from the Holocene onset to present day. This discrepancy may be due to the low resolution of our model, as the Renland ice cap covers only 1200 km$^2$ (Johnsen et al., 1992), and the model cannot capture the steep descents in topography that limit its lateral extent.

## 6 Conclusions

We considered an ensemble of ice-sheet model simulations covering both Greenland and the Canadian Arctic Archipelago through the Holocene. In these simulations, we varied 20 key parameters to constrain the ice sheet evolution to ice-core-derived surface elevation histories at four ice-core sites in Greenland. We showed that the inclusion of Canada in the model domain and the ability of the ice sheet to advance beyond the present-day land margin are necessary for accurately modeling the ice-core-derived elevation history.

We found that during the last glacial period, the GrIS was connected to the IIS with an ice bridge over Nares Strait. Within the ECS, the GrIS had an extent that was 49% larger than the present-day modeled area 12 ka ago, and it was found to have contributed 5.3±0.3 m SLE to the global mean sea level from 12 ka ago to present day. The collapse of the ice bridge at Nares Strait was found to have occurred at 4.9±0.5 ka before the present.

We show that the present mass loss rate is a combined short-term response to the recent climate forcing and long-term dynamic response on millennia timescales due to the deglaciation history. Ignoring outliers with excessive temperature anomalies over the past half-millennium, we find that the mass loss rates over the last 500 years primarily depend on the timing of the onset of ocean forcing during the deglaciation. Bayesian inference modifies our understanding of the previous 500 years' mass loss from a prior estimation of 12±40 mm ka$^{-1}$ to a posterior of 23±26 mm ka$^{-1}$, which is about 5% of the 1992-2020 estimated mass loss rate (The IMBIE Team, 2020) and 7% of the estimated 21st-century committed mass loss rate (Nias et al., 2023). This adjustment underscores the significance of paleo calibration in accurately modeling ice sheet behavior and including its long-term response to past climatic changes.

While our study was able to model the ice-core derived surface elevation histories, the most probable ice sheet simulations did not match the timing of the ice bridge collapse found by England et al. (2006) or the timing of the retreat found by Leger et al. (2024). We propose that these geologically derived datings could be added as further constraints to the GrIS Holocene evolution in future simulations. This would help reveal limitations in the model and assess the sensitivity of model parameters. We also found that our modeled present-day uplift rates deviated from the GPS-derived uplift rates in northwest Greenland,

which is further in line with the timing of our model collapse happening too late. In future studies, these deviations should be used to constrain the mantle viscosity in tandem with accurately determining the timing of the retreat. Overall, our results show that the present-day GrIS still responds to the history of deglaciation. This long-term dynamic response is significant and
485 should be included in studies of the present and future mass loss from the GrIS.

*Code and data availability.* PISM is an open source software that can be downloaded from github.com/pism/pism (Bueler and Brown, 2009; Winkelmann et al., 2011). Surface elevation data from the four ice core locations are available upon request. The presented RACMO data are available upon request and without conditions from B. Noël (bnoel@uliege.be). Temperature reconstructions can be downloaded from iceandclimate.nbi.ku.dk/data/.

*Video supplement.* A video showing the GrIS evolution through the Holocene for the most likely ensemble member can be found at doi.org/10.5446/68337

## Appendix A

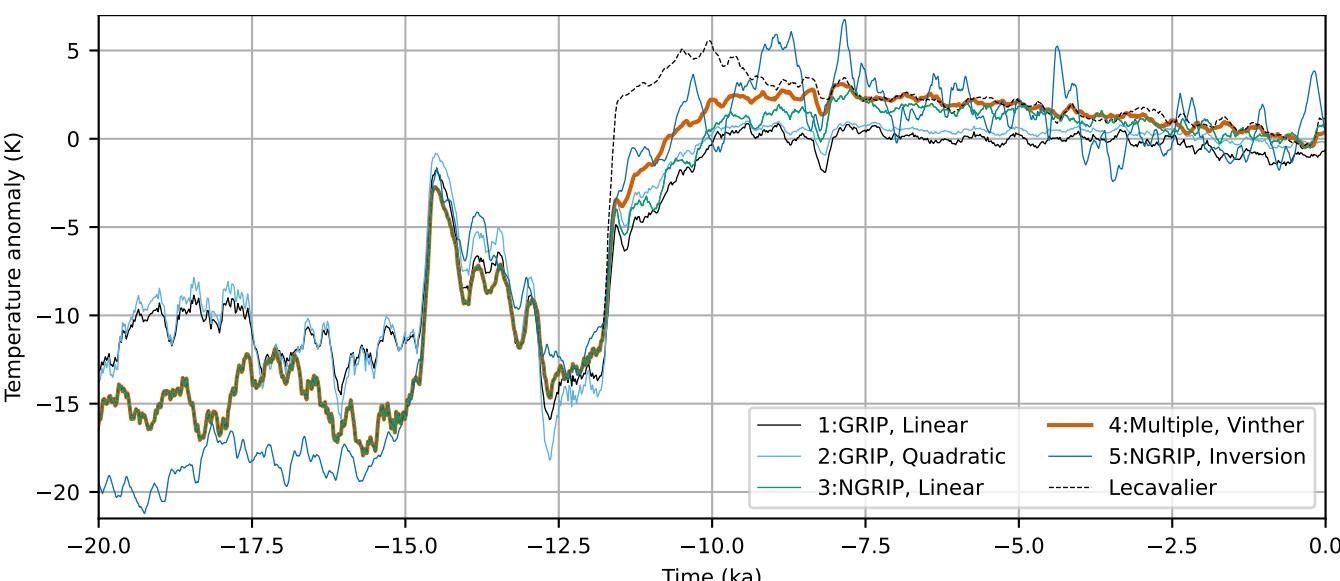

**Figure A1.** Paleoclimatic temperature anomalies derived from O-18 measurements at GRIP and NGRIP using linear transfer function (Huybrechts, 2002) and quadratic transfer function from Johnsen et al. (1995) and O-18 measurements at Renland and Agassiz (Vinther et al., 2009) and O-18 measurements at NGRIP using an inversion scheme (Gkinis et al., 2014).

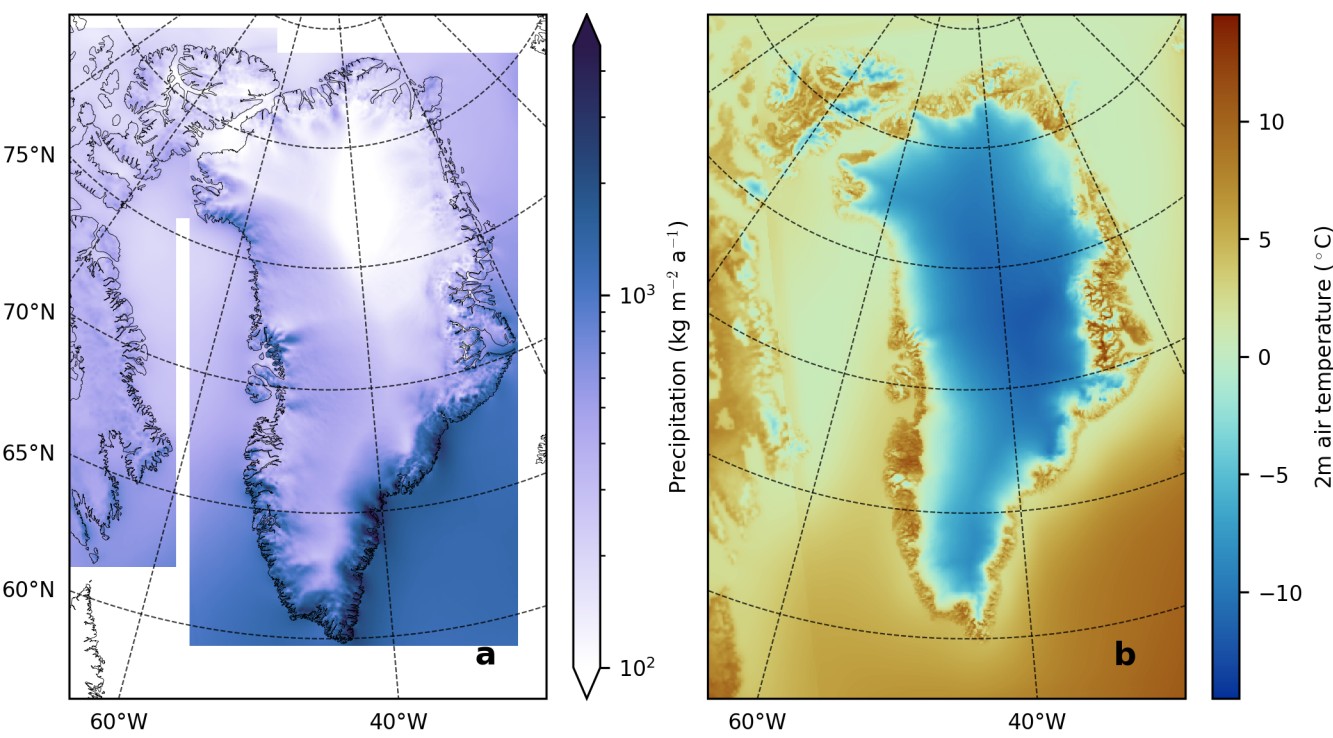

**Figure A2.** (a) Annual mean precipitation and (b) summer (June, July, and August) mean 2 m temperatures for our 30-year reference climatology (1960-1989).

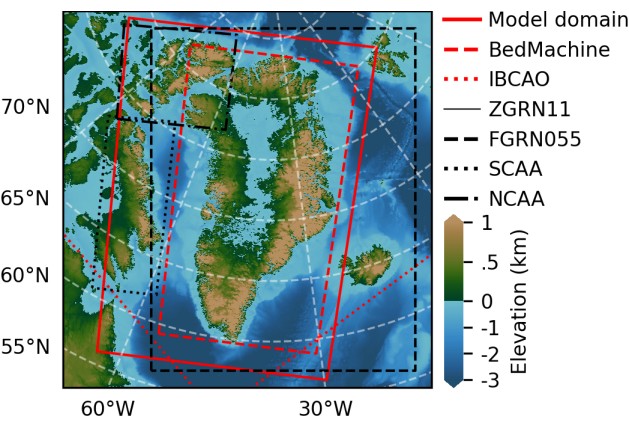

**Figure A3.** Overview of different domain boundaries used to patch together the 30-year reference climatology (ZGRN11, FGRN055, SCAA, NCAA) and the bedrock topography (BedMachine, IBCAO, GEBCO).

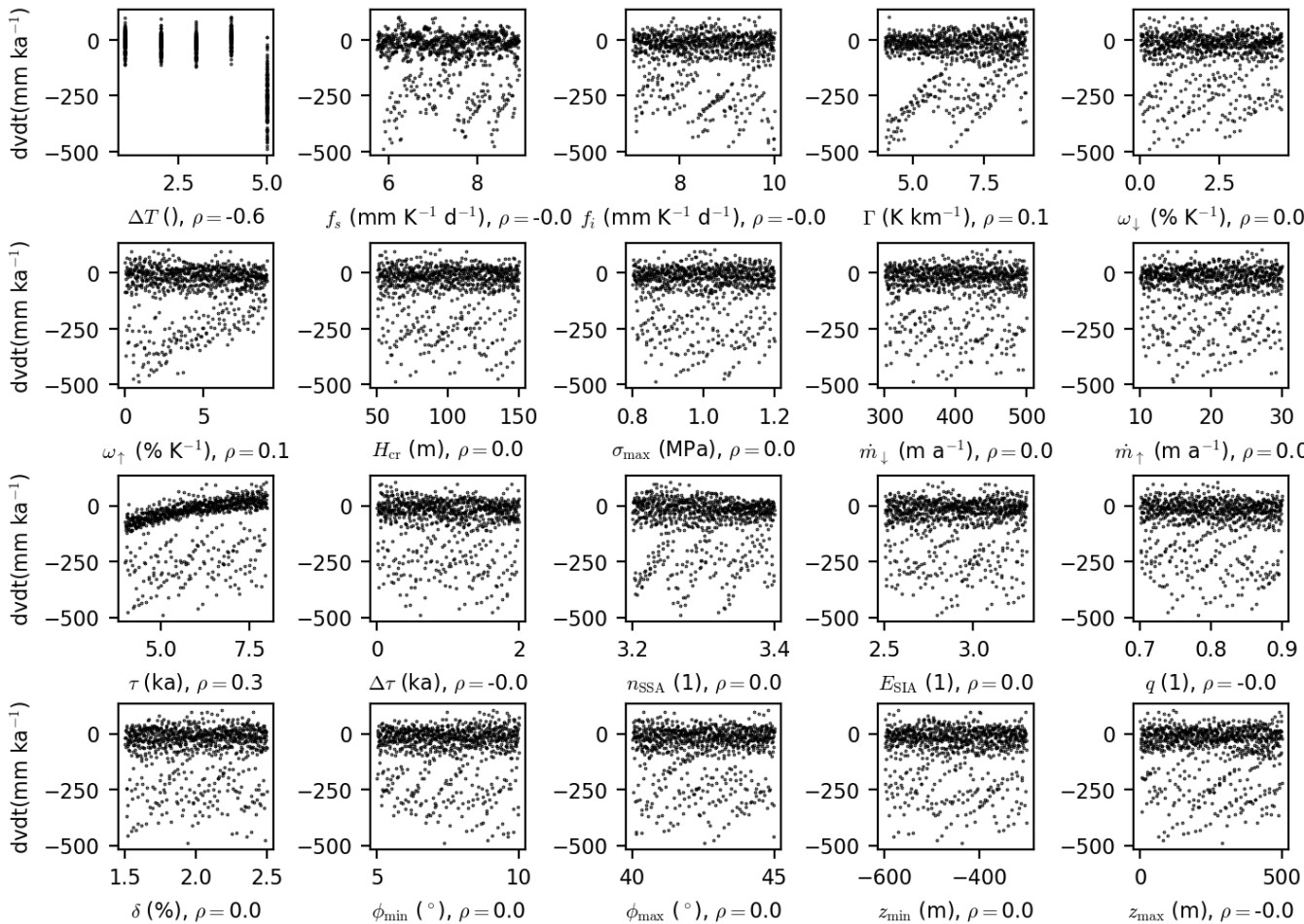

**Figure A4.** Scatterplot of the last 500 years of mass loss rates vs each parameter varied in our ensemble. $\rho$ is the Pearson correlation.

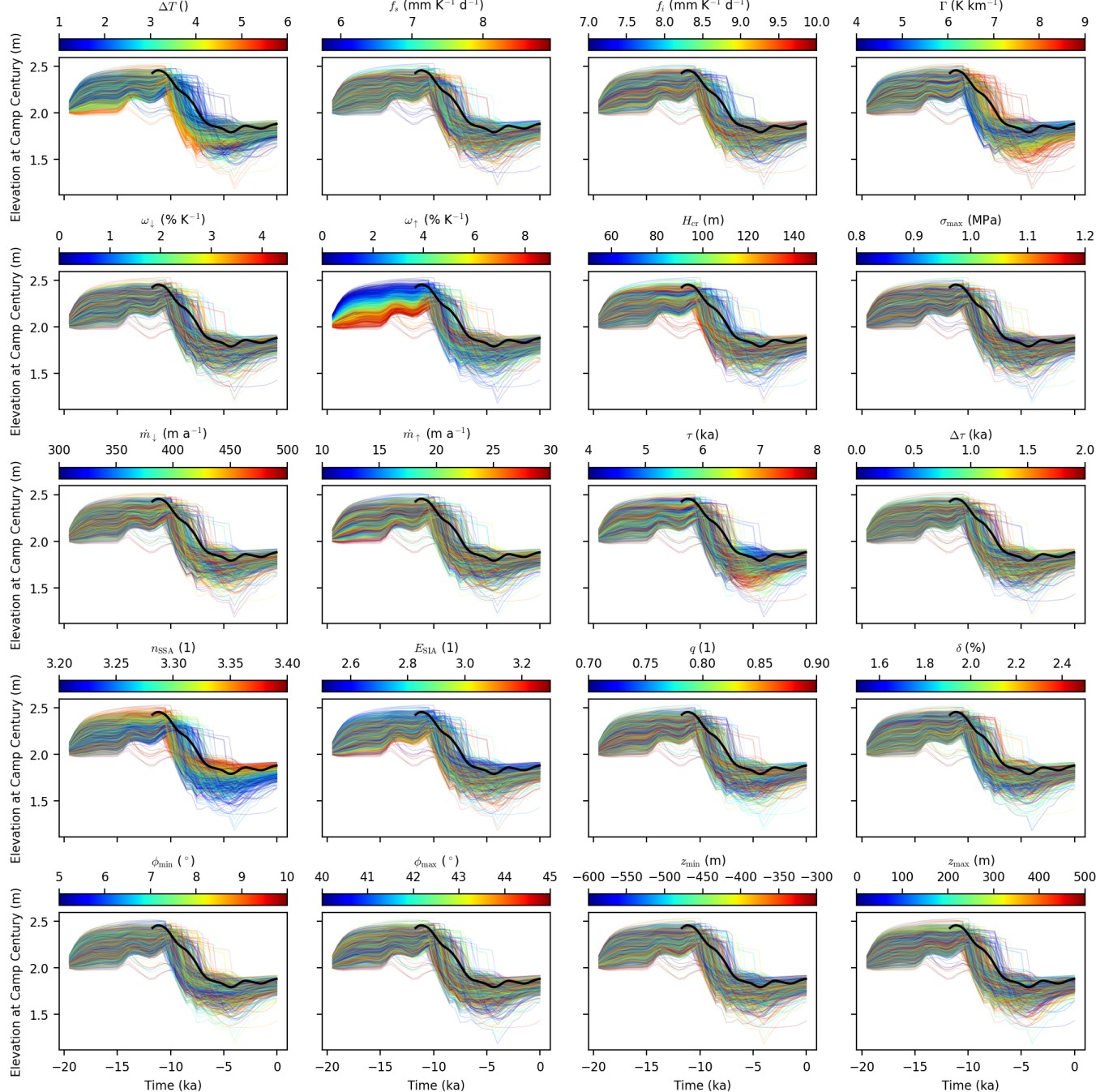

**Figure A5.** Modeled surface elevation at Camp Century color coded for each parameter.

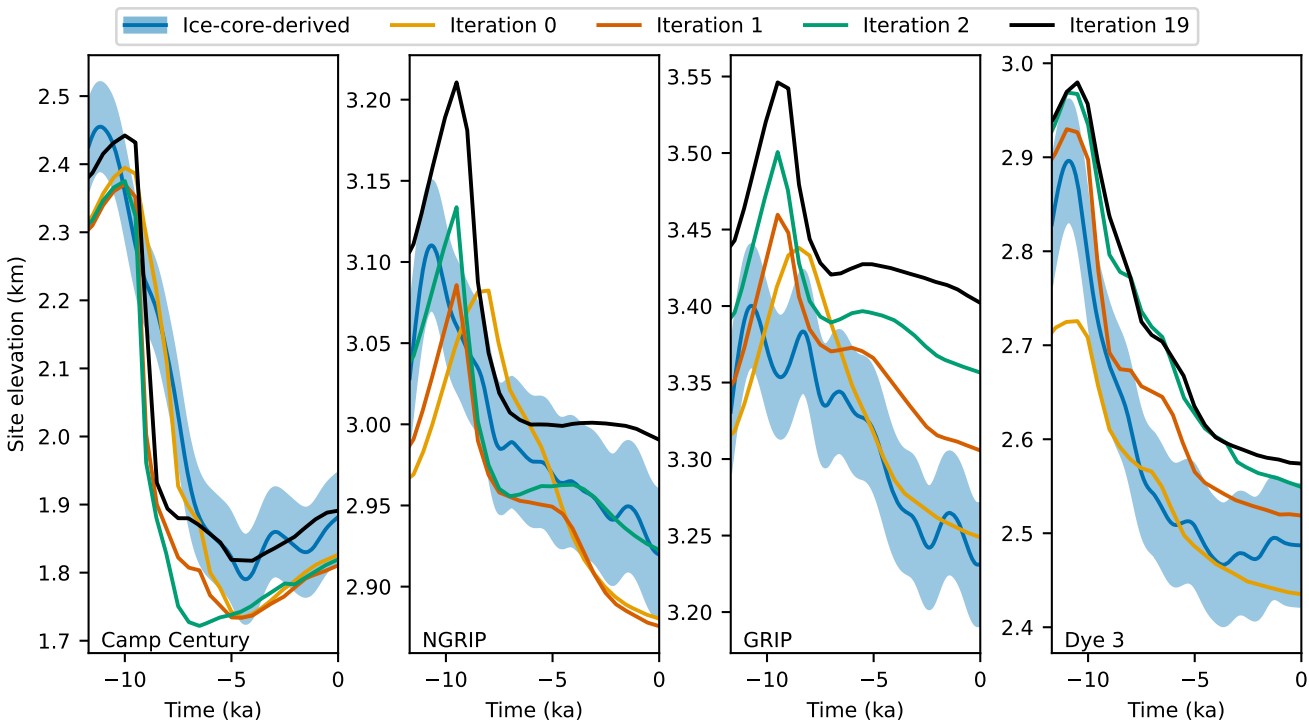

**Figure A6.** Observed and modeled surface elevation over the past 11.7 ka at the ice-core sites Camp Century, NGRIP, GRIP, and Dye 3. The blue lines represent ice-core-derived surface elevations from Vinther et al. (2009), with the blue envelopes indicating one standard deviation. The orange, red, green, and black lines correspond to the modeled surface elevations for the 0th, 1st, 2nd, and 19th iterations of the bedrock adjustment, respectively.

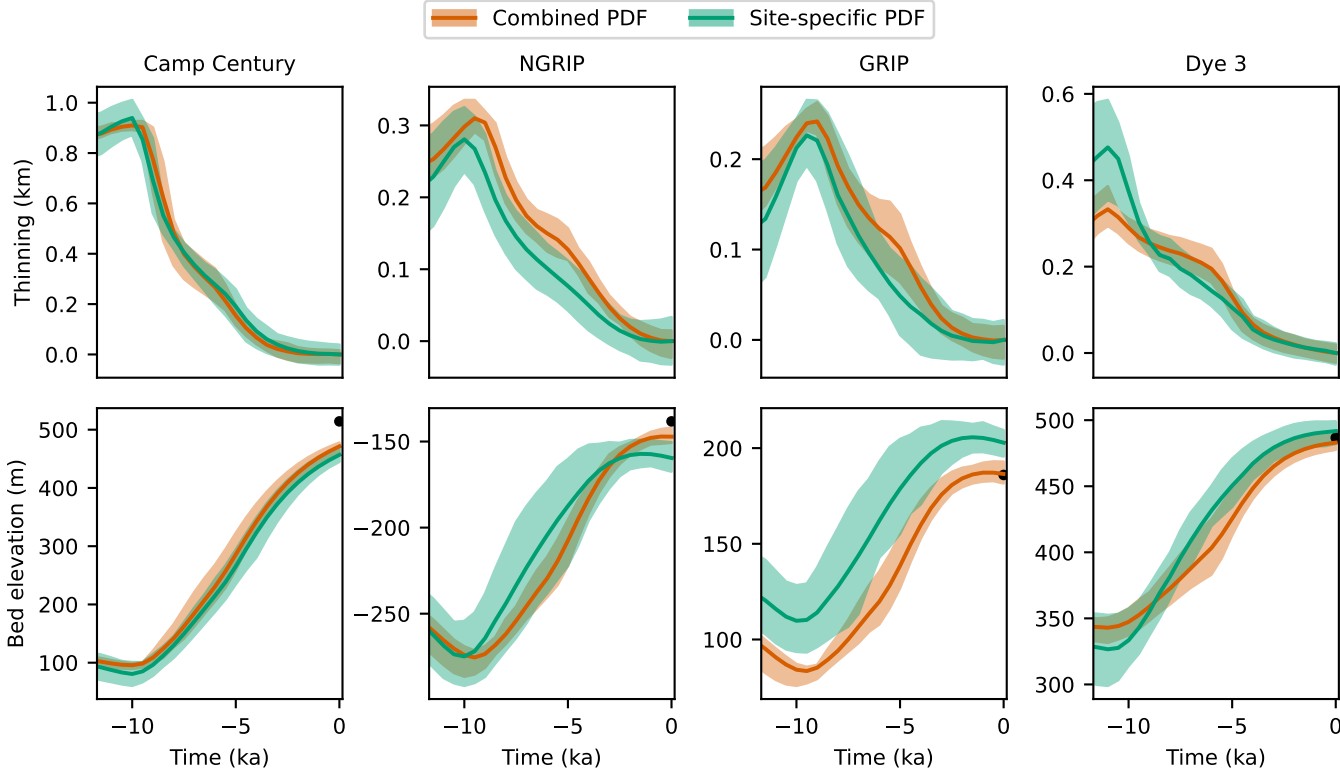

**Figure A7.** Modeled bedrock elevation and thinning over the past 11.7 ka at the ice-core sites Camp Century, NGRIP, GRIP, and Dye 3. The red envelopes represent the ensemble-estimated mean and standard deviation, while the green envelopes show the site-specific estimate. The black dots indicate the observed present-day bedrock elevation from Morlighem et al. (2017).

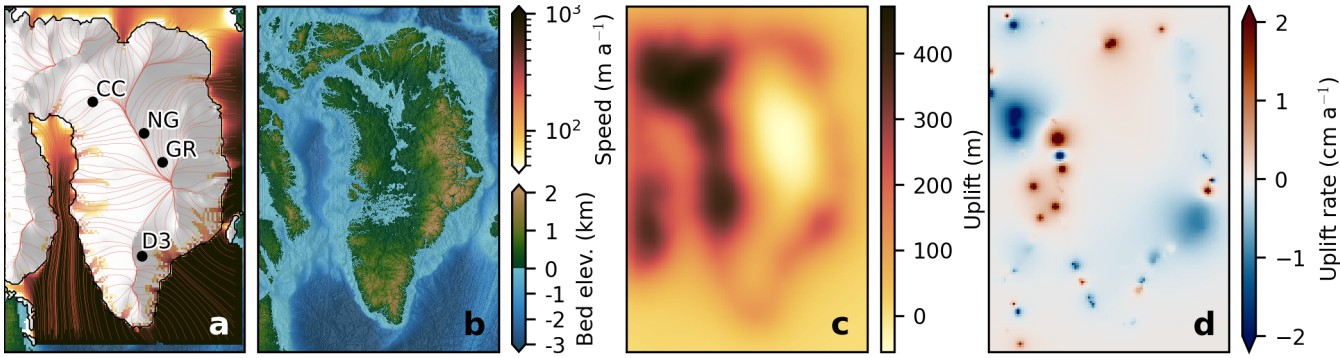

**Figure A8.** Model state at the branch-off point at -20 ka. (a) Modeled surface velocity with streamlines and ice shelf extent. The present-day locations of the ice-core sites Camp Century (CC), NGRIP (NG), GRIP (GR), and Dye 3 (D3) are overlaid. (b) Bedrock topography at -20 ka relative to sea level. (c) Difference in bedrock topography at -20 ka compared to the present day observed ($b_{PD} - b_{20ka}$). (d) Modeled bedrock uplift rates at -20 ka.

*Author contributions.* ML, CH, and AS designed the study. ML prepared the data, performed the model runs, and carried out the subsequent analysis. All authors discussed and improved the paper.

*Competing interests.* The authors declare that they have no conflict of interest.

*Acknowledgements.* ML was funded by the Independent Research Fund Denmark through the project GreenPlanning, grant no. 0217-00244B. CH, NR, and AG received funding from the Novo Nordisk Foundation, grant no. NNF23OC0081251, the Independent Research Fund Denmark (DFF), grant no. 2032-00364B, and the Villum Foundation, grant no. 23261. BN is a Research Associate of the Fonds de la Recherche Scientifique de Belgique – F.R.S.-FNRS. AS was funded by the European Union (ERC, Green2Ice, 101072180). Views and
500 opinions expressed are however those of the author(s) only and do not necessarily reflect those of the European Union or the European Research Council Executive Agency. Neither the European Union nor the granting authority can be held responsible for them. We would also like to thank Benoit Lecavalier for providing the temperature anomalies derived from the Agassiz ice core in Lecavalier et al. (2017). We would like to thank the two anonymous reviewers and the editor, Alexander Robinson, for their help in improving the manuscript.

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
