# Peer review of "Modeled Greenland Ice Sheet evolution constrained by ice-core-derived Holocene elevation histories"

_EGUsphere, 2024_

## Author Response (AR1)

**Modeled Greenland Ice Sheet evolution constrained by ice-core-derived Holocene elevation histories**

Addressed Comments for Publication to

**The Cryosphere**

by

Mikkel L. Lauritzen, Anne Solgaard, Nicholas Rathmann, Bo M. Vinther, Aslak Grindsted, Brice Noël, Guðfinna Aðalgeirsdóttir, Christine S. Hvidberg

Dear Alexander Robinson,

Please find enclosed our responses to the submission entitled "Modeled Greenland Ice Sheet evolution constrained by ice-core-derived Holocene elevation histories". We would like to thank you and the reviewers for the valuable comments which helped improving the quality of our manuscript. We have carefully addressed the reviewers' comments. A detailed point-by-point response to the comments from Referee 1 and 2 and the three community comments are given below.

Sincerely,

Mikkel L. Lauritzen, Anne Solgaard, Nicholas Rathmann, Bo M. Vinther, Aslak Grindsted, Brice Noël, Guðfinna Aðalgeirsdóttir, Christine S. Hvidberg

**Authors' Response to Referee 1**

**General Comments.** This work investigates the evolution of both the Greenland Ice Sheet (GrIS) and the Canadian Arctic Archipelago during the last deglaciation using simulations of the PISM ice-sheet model. It explores how Holocene surface elevation histories, derived from ice-core data across the ice sheet, can be used to validate model results and refine model parameters. A well-known issue in the field is the difficulty of 3D ice sheet models in accurately simulating Holocene thinning curves across Greenland: this work successfully replicates these curves and attributes the previous data-model discrepancies to the limited modeled ice-sheet advance and the lack of a connection between the Innuitian ice sheet (IIS) and the GrIS during the Last Glacial Maximum (LGM).

Given the scarcity of information on the GrIS evolution away from the margin, surface elevation histories can be an additional useful metric for model validation helping to constrain the evolution of the ice-sheet interior and the overall volume. By considering a domain that includes the IIS and by adjusting the bedrock elevation at the LGM, the authors can replicate the Holocene thinning curves with good agreement upon having calibrated 20 key model parameters.

I find this a very interesting work. It shows how the Holocene ice-core derived surface elevation changes can be used to validate the modeled ice-sheet evolution, as long as the response of the GrIS to non-local changes in the ice load and/or ice dynamics are considered. However, given the high uncertainty in deriving such thinning curves from ice cores and the availability of both more reliable paleodata sources, such as exposure dates of moraines, and present day observations, I think one must be careful in using these elevation histories as - almost - the only metric for validating model simulations. I strongly agree with Jessica Badgeley, who commented in the discussion. While I won't reiterate her points, I believe she raised several important considerations. Additionally, there is a significant discrepancy between the modeled retreat history and that reconstructed from proxy data, particularly regarding the timing of the retreat. This issue needs to be addressed before considering the elevation history of Greenland, as the magnitude and timing of bedrock uplift and ice thickness changes are influenced by the timing of ice margin retreat.

**Response:** Thank you for your feedback and interest in our work.

We have carefully addressed all the issues item by item as follows.

**Comment 1**

Wrong timing and spatial variability of ice-sheet retreat: The GrIS evolution simulated throughout the deglaciation does not match the timing of the ice margin retreat inferred from several proxies (most of all moraines) and geomorphological reconstructions. Leger et al. 2023 did an amazing work in collecting and processing most of data available across Greenland to provide a detailed information on area change and timing of the deglaciation across the ice sheet. Most of the GrIS deglaciated already before 9 kyr ago (see Figure 15 of Leger et al., 2023) while your figure 6b still shows a glacial-expanded Greenland. This suggests that simulations presented here are systematically late in modeling the ice-sheet shrinkage during the last deglaciation and I am wondering how this might affect the simulated ice-core thinning curves. I think this is a key aspect that has to be solved before investigating the thinning curves in detail, otherwise you might find a good elevation history match but for the wrong reasons. Surface elevation changes occur because of changes in bedrock elevation, but also because of changes in ice thickness, which are in turn related to changes in the mass balance and ice dynamics. Ice dynamics and local changes in bedrock elevation are associated with the retreat history of Greenland, the former mostly due to dynamic reorganization occurring during the deglaciation, and the latter to changes in the ice load. I do believe that simulating a correct bedrock elevation change during the last deglaciation is key to correctly replicate the Holocene surface elevation change. However we should first be able to simulate the right retreat history of Greenland before trying to replicate the Holocene surface elevation curves from ice cores with a model.

**Response:** Thank you for the comment.

The simulations presented here are indeed systematically late in modeling the ice sheet retreat during the last deglaciation and the ice bridge between the IIS and the GrIS collapses about 3 ka later than the findings by England et al. (2006) as pointed out in line 272-274.

We agree that modeling the timing of the retreat would improve the overall confidence of the simulations, provided that the ice dynamics and basal hydrology and sliding are realistically represented in the model. In reality, however, the ice flow models have limitations, and constraining the evolution to just the retreat also risks getting the retreat right for the wrong reasons. Of course the marginal retreat depends largely on the ocean forcing but also, as you mention, on the ice dynamics and the bedrock elevation histories. Our approach in this paper is to focus on one aspect, the ice core derived surface elevation history, as a new and independent

first step, as it provides important new insights to the limitations of the ice flow model. Clearly the next step, for a further study, would be to use also retreat rates as constraints. As discussed in lines 274-277 the ensemble members with earlier onsets of oceanic forcing becomes too thin at Camp Century in the middle of the Holocene. This suggests that further work is needed to understand this apparent inconsistency between the proxies.

> In the revised version of the paper we have provided an additional figure (Fig. A6) that shows the isochrones for the modeled retreat and the moraine-line-derived retreat of Leger et al. (2024).
>
> We discuss how the margin retreat can be used in the next step in the model development in section 6.1. In the revised paper we have also provided a new figure (Fig. A5) that highlights the mismatch in elevation at CC during mid-Holocene for the ensemble members which are color coded for each parameter.

**Comment 2**

Importance of bedrock adjustment: I find the procedure for adjusting the bedrock quite interesting but I think I don't fully understand the motivation/significance of it. What does this strategy mean conceptually: why do you need to refine the bedrock at the LGM, if the model simulates the glaciostatic adjustment? Is this a necessary step to well replicate the modeled surface elevation history throughout the Holocene or is it a strategy to reduce the elevation mismatch only at the present? Have you run an ensemble of simulations without such an adjustment to evaluate its effect on the ice-sheet shrinkage during the last deglaciation? I think that a more detailed justification of this procedure and a discussion about the implications of such a procedure on your results is needed.

**Response:** Thank you for the comment.

We only know the bedrock at present day, not in the past. The model simulates the glaciostatic adjustment over time from an unknown state at LGM, and when the model does not match the present day bedrock at the end of the run, we need to refine the initial bedrock at -20 ka, and re-run the model. When running model without adjusting the initial bedrock at -20 ka, the glaciostatic adjustment does not return the bedrock to the observed state at present as shown in Fig. 3b. Changing the bedrock does impact the elevation history, but we would not consider

it essential for modeling the ice-core-derived elevation histories. However, using an incorrect bedrock configuration could produce the correct elevation histories for the wrong reasons.

> In the revised paper we have clarified the procedure in section 3.5 and provided a figure (Fig. A7) showing the effect of bedrock adjustment in the model simulations on the surface elevation histories as well.

**Comment 3**

Outdated paleo climatology: using a spatially homogeneous temperature and precipitation signal to force the model during the last deglaciation is an outdated procedure. There are several products available with high temporal resolution (e.g. Badgeley et al, 2020, Buizert et al., 2018), based on the trace21k experiment (Liu et al., 2009, He et al., 2013) and improved with ice-core derived information that can do the job better. These paleoclimate reanalysis have enhanced our knowledge on the Holocene climate in Greenland and make the anomaly method for this period completely outdated. These products have already been used in several works tackling the GrIS evolution during the last deglaciation (Briner et al., 2020, Cuzzone et al., 2022, very recently in Tabone et al, 2024) and they clearly show that considering a spatially homogeneous temperature across Greenland is incorrect. I believe that using such data would also help to reproduce the spatially variable retreat suggested for the last deglaciation (Leger et al., 2023).

**Response:** Thank you for the comment.

We agree that using a spatially homogeneous temperature is an old procedure. However, this choice was firstly made to align with the assumption of Vinther et al. (2009) that local temperature offsets were due to a Greenland wide temperature anomaly and local elevation feedback. Secondly, we chose the five different and frequently used temperature reconstructions and sampled over a range of atmospheric lapse rates to cover the historic temperature uncertainty. This eased the interpretation of the Bayesian inference and enabled us to tell if there was any regional discrepancies as pointed out in lines 215-220.

The TraCE-21K climate simulations use the ice sheet configuration of ICE-5G which is not consistent with our simulated ice sheet and does not include an ice shelf covering the Baffin Bay. Placing too high confidence in these products could introduce a different bias and when making

a reconstruction of the Holocene evolution we prefer to keep it independent of any previous reconstructions to avoid any circular conclusions.

The precipitation field is not spatially uniform as detailed in lines 118-121 and eq. 1. We introduce a meridional gradient in the accumulation scaling exactly because we want to allow for different accumulation histories in the north and the south.

> In the revised paper we have included a discussion of the use of non-uniform temperature anomalies and reanalysis products, and added a reference to the work of Badgeley et al, 2020 (section 6.7).

**Comment 4**

Discussion missing on drivers of Holocene thinning: I am missing a clear discussion on what is driving the Greenland elevation change following your simulations. You claim that by including the IIS and an ice shelf in Baffin Bay you can reproduce the Holocene elevation history in Greenland, but which is the glaciological explanation for it? Is it the bedrock response to non-local changes in the ice load, or is it the dynamic effect induced by the loss in buttressing upon ice shelf retreat in Baffin Bay, or both of them or what is responsible for the Holocene elevation drop in Greenland? If the loss of buttressing is the preponderant mechanism, why do we see a drop in elevation already several thousand of years prior to the ice shelf collapse? Please, provide a clearer explanation of your findings as this is key to improve our knowledge in the field.

**Response:** Thank you for the comment.

It is the changes in climate conditions that are driving the change in mass balance which in turn changes the dynamics of the ice sheet and forces it to thin. The bedrock adjusts to the load and generally acts as a negative feedback to the changes. The ice sheet starts to thin at the margin as a response to the temperature rise at the early Holocene which leads to thinning in the interior w/ and w/o bedrock adjustment. This also leads to the collapse of the ice shelf in Baffin Bay which further reduces the buttressing. After the onset of the ocean forcing, the mass loss speeds up again as seen in Fig. 8 and the ice bridge at Nares Strait collapses. We need the ice sheet to advance beyond the present-day boundaries in order for the ice sheet to be thick enough at the four ice core sites at the LGM as shown by the two simulations restricted to the present-day land margin and the ECS. While it might be possible to get an ice sheet profile

that matches both the present-day extent and the LGM thickness at the four ice core sites by changing the viscosity spatially, this would be modeling the right elevation for the wrong reason, as we know the ice sheet extended beyond the present-day extent.

> In the revised version of the paper we have added to the discussion of the drivers of Holocene thinning (section 6.8) and a motivation for the necessity of including the extended ice sheet (section 6.1).

**Comment 5**

Usage of other present-day observables: I understand that the estimation of model uncertainty/model parameter calibration has been done by applying a Bayesian approach only to the ice-core surface elevation histories, whilst the others "observables" of Table 2 (present day thickness, surface velocity) are estimated from the resulting probability density functions. Why not including such "observables" already in the bayesian inference to validate your model simulations? This would allow to not rely only on surface elevation change (which is much more uncertain than satellite inferred observables) to evaluate model results.

**Response:** Thank you for the comment.

The reason for excluding present-day observables from our Bayesian inference is that we are primarily interested in testing the feasibility of using the past surface elevation history to understand how prognostic (future) simulations might be biased by the long-term response of the GrIS to past climatic changes. To judge this, we argue that using this data in isolation makes our results most clear. We can see that this important goal of assessing the long-term response of the GrIS is not highlighted sufficiently in the paper.

Present-day observables are included only as a source of independent data for model validation. Our goal is not to replicate the present-day state within the bounds of observational uncertainty, which is, to the best of our understanding, a more generally unresolved problem.

> In the revised version of the paper we argue why we only use the ice-core-derived surface elevation histories in our Bayesian inference (section 4). Further, we have clarified in the introduction that our goal is to assess the long-term response of the GrIS to past climatic changes.

**Comment 6**

Unclear glacial spin up: Is the model run with a fixed LGM climatology for 80 kyrs or is it run forced with a temporal variable temperature and precipitation signal? If it is the latter, which is the glacial climatology used? Moreover, how are the parameters set up during the spin up? If I understand correctly, only one spin up is done for all the simulations in the ensemble, then the response of the model to the 20 key parameters is explored only since the LGM. If this is so, this might lead to a certain "shock" of the model in the first years after the initialization (after the LGM). Actually, I think I see this shock in Figure 7b, where the volume suddenly drops after the LGM for the "constrained to present geometry" case. I don't think this procedure undermines your results, since your work focuses on the Holocene (which starts after 8000 years from this "new" initialization), but it would be good either to do one spin up for each ensemble member to avoid this inconsistency or at least describe the first years of your ensemble simulations as part of the spinup.

**Response:** Thank you for the comment.

The glacial spin-up is indeed done using the temperature anomaly record based on the NGRIP ice core (reconstruction number 3) and by using a single set of parameters. We will make sure to state these.

Yes, there is indeed only one spinup which is then branched at -20 ka to ensure that the ensemble members have enough time to recover from the shock of changing the bed topography and the parameters.

> In the revised version of the paper the parameters and temperature forcings used for the spin-up are listed in table 1. We have described the branching of the ensemble members between -20 ka and the onset of the Holocene at -11.7 ka in section 5.2.

Specific comments:

**Comment 7**

Please check the citation format throughout the text: The sentence "as demonstrated by (Adalgeirsdóttir et al., 2014)" should be "as demonstrated by Adalgeirsdóttir et al. (2014)", for instance.

**Response:** Thank you for the comment.

This is indeed a blunder. We have changed the citation format and made sure that the rest are correct.

> In the revised version of the paper the citation format is consistent.
* * *
**Comment 8**

Lines 6-8: How can we have "confidence in the modeled GrIS historical evolution" if the model does not reproduce the retreat history correctly?

**Response:** Thank you for the comment.

That is a good question. We will have greater confidence in the simulations that are constrained to the Holocene surface elevation histories than simulations that are not. That being said, reproducing the retreat history would further enhance our confidence of the simulations and the long term response. As written in the reply to comment 1, we believe that reproducing the internal thinning is an important first step to increase our confidence.

> In the revised version of the paper we have provided an additional figure (Fig. A6) that shows the isochrones for the modeled retreat and the moraine-line-derived retreat of Leger et al. (2024).
>
> We discuss how the margin retreat can be used in the next step in the model development in section 6.1. In the revised paper we have also provided a new figure (Fig. A5) that highlights the mismatch in elevation at CC during mid-Holocene for the ensemble members which are color coded for each parameter.
* * *
**Comment 9**

Lines 38-41: this is not entirely true. What about the constraints on past ice-sheet extent given by e.g. moraines, marine sediment cores, coastal organic material? This is a reliable information we do have on past GrIS retreat.

**Response:** Thank you for the comment.

By no direct measurements we mean that there are no satellite observations.

> In the revised paper the sentence in line 38-41 has been rephrased.

**Comment 10**

Lines 44-45: I think here a more detailed description about previous work is needed. It is true that previous work did not consider the effect of the IIS on the deglaciation but it was still successful in reducing the data-model misfit at specific ice-core sites. Lecavalier et al., 2017 was actually successful in reproducing the magnitude of the Camp Century elevation change by correcting the Holocene climate forcing in North Greenland, as reconstructed from the Agassiz record, to force an ice-sheet model coupled to a GIA model of relative sea level change. Another very recent work (Tabone et al., 2024) was able to reduce the mismatch at the NGRIP site by considering the effect of the paleo NEGIS dynamics on the ice-sheet interior using a 3D ice-sheet model.

**Response:** Thank you for the comment.

Yes, Lecavalier et al. (2017) were able to reproduce the thinning at CC without including the effect of IIS. We mention this in lines 362-367, but will move it to lines 44-45.

> In the revised paper the introduction includes a more detailed description on the previous work to model GrIS Holocene elevation changes including the work of Lecavalier et al. (2017) and Tabone et al. (2024).

**Comment 11**

Lines 60-70, 114-116: as written in the general comments, using a homogeneous temperature reconstruction from ice cores to force the model is unnecessary here, as there are already several deglaciation climate products (Badgeley et al., 2020, Buizert et al., 2018) which do consider the spatial climatic variability that the GrIS experienced during the last deglaciation. I think using this reanalysis instead of a uniform climatology would allow to better replicate the asynchronous ice-sheet retreat during the last deglaciation (see below and Leger et al., 2023).

**Response:** Thank you for the comment.

As written in the previous response, we choose to use the spatially uniform temperature reconstructions to make use of Vinther et al. (2009) data in most logical way. As a next step it might be a good idea to use non-uniform temperature reconstruction although we believe that keeping the Holocene reconstruction of the Greenland Ice Sheet independent of any previous reconstruction would be best to avoid circular conclusions.

> In the revised paper a discussion on the use of uniform temperature anomalies has been included (section 6.7).

**Comment 12**

Lines 130-131 and Table 1: ocean melt onset parameter "tau": why is this explored between -4 and -8 ka, whilst Clark et al., 2020 show an increase in the oceanic forcing already at the early Holocene (at 45°N, 30°W, figure 1k)? Below in the manuscript it is found that the sub-shelf ocean melt that best matches the Camp Century thinning curve starts around -5 ka. Is there any evidence of this, e.g. from sediment cores in Nares Strait?

**Response:** Thank you for the comment.

The ocean onset parameter, tau, is explored between -4 and -8 ka as this is the interval showing the highest ice-core-derived elevation likelihood after doing a few exploratory simulations. There is no evidence of ocean melt increase from sediment cores at this time to our knowledge. The time is only what is inferred from the Bayesian calibration when constrained to the ice-core-derived elevation changes.

> In the revised version of the paper we state that the parameters ranges are chosen by the volume showing the highest likelihood after an initial ensemble of simulations.

**Comment 13**

Lines 155-156: could you show a 2D map of the bedrock at -20 ka after it has been adjusted? Also, from Figure 2 it seems that the adjusted LGM bedrock elevation is higher than that of the present day, on average, but I believe that it is lower than the one simulated at the LGM before the iteration. . .

**Response:** Thank you for the comment.

This is a good suggestion that we would like to act on by creating a new figure to be included in the appendix. Figure 2 is a schematic figure that shows the different steps of the experiment and does not show the elevation. While the x-axis depicts time the y-axis is only used to reflect the fact that the states differ.

> In the revised version of the paper we have included a figure to show the bedrock topography at -20 ka (Fig A9).
>
> In the revised version of the paper we have clarified the caption of Fig. 2.

**Comment 14**

Line 162: "unwanted deglaciation": is this because by lowering the bedrock elevation at the margin (I guess without updating the ice thickness), the surface elevation decreases too, therefore the ice sheet surface becomes exposed to higher air temperatures?

**Response:** Thank you for the comment.

The idea behind the relaxation parameter was that there might be some positive feedback that would add to the imposed changes in the bedrock adjustment. However, it is not clear if this feedback exists as the bedrock-surface-mass balance feedback is negative and therefore the relaxation parameter might be redundant although we do not explore this.

> In the revised version of the paper we have added a sentence saying why we introduced the relaxation parameter in section 3.5.

**Comment 15**

Figure 3: why not showing the 2D bedrock elevation at the last iteration too, for comparison?

**Response:** Thank you for the comment.

This is a good idea that would make the comparison easier for readers, the last iteration 2D bedrock elevation will replace the third one that is now in Figure 3.

> Figure 3 have been revised to show the bedrock elevation at the last iteration rather than the 3rd.

**Comment 16**

Line 169: "while the oceanic and atmospheric parameters chosen reflect the change in our model setup": unclear, please rephrase.

**Response:** Thank you for the comment.

We agree that the sentence is unclear and will rephrase it in the revised version of the paper. We meant to say that the dynamic parameters are the same as those varied by Aschwanden and Brinkerhoff (2022), while the oceanic and atmospheric parameters are the ones that are introduced in the simulations presented in this paper.

> In the revised version of the paper sentence in line 169 has been rephrased.

**Comment 17**

Line 202: "two simulations were restricted from advancing beyond the present-day GrIS coast...". To my understanding one simulation did not advance from the present-day GrIS coast, and the other from the Greenlandic continental shelf, isn't that it? Please rephrase.

**Response:** Thank you for the comment.

Yes, that is indeed correct. We will rephrase this.

> In the revised version of the paper sentence in line 202 has been rephrased.

**Comment 18**

Lines 202-206: As far as I understand, you initialize the model at the LGM prior to start the simulation ensemble, so how can it be that you simulate different GrIS extents at the end of the glacial period depending on the parameters set? I would expect that all your simulations show a well advanced LGM, since this configuration is generated by the same initialization, and then in the ensemble you explore different deglaciation patterns. What am I missing?

**Response:** Thank you for the comment.

Yes, all simulations start from the same state at -20 ka. One simulation is then forced to retreat to the ECS mask and one to the present-day land mask by removing all ice outside their respective masks.

> In the revised paper the sentence in lines 202-206 has been revised to make it clear that the two simulations are only restricted after the spin-up.

**Comment 19**

Lines 230-231: what about the configuration at the LGM? Why do you simulate a maximum glacial extent only at -12 ka? Can this be a drift of the model as it might still be adapting to the new parameters set after the initialization at the LGM?

**Response:** Thank you for the comment.

This is a good point, there might be a drift in the model due to adaptation to the new parameter selection and bedrock adjustment. It is confusing to use the model state at -12 ka to resemble the LGM, this will not be done in the revised version of the paper.

> In the revised version of the paper we have described the model shock to the new parameters.

**Comment 20**

Lines 243-244: "The GrIS rate of change becomes negative at -10.7 ka and peaks at -7.8 ka with a mass loss rate of 548 Gt a-1", is it? Or does it peak around -5 ka? I don't see such a mass loss rate either.

**Response:** Thank you for the comment.

This sentence is admittedly a bit unclear. There are two peaks of mass loss rate, one at -7.8 ka and one at -4.95 ka with mass loss rates of 548 and 511 Gt a-1 respectively.

> In the revised version of paper the sentence on lines 243-244 have been rephrased to make it more clear. We have also changed "4.95 ka" to "-4.95 ka" and made sure that this is consistent.

**Comment 21**

Figure 6: please change the color scales so that the bathymetry can't be confused with the surface velocity. Also, the "bridge" between the LIS and the GrIS is floating ice, right? Please show clearly where are the ice shelves in these maps.

**Response:** Thank you for the comment.

We agree that the color maps may result in a risk of confusion in interpreting the figure. No, the bridge between the IIS and the GrIS is grounded while the ice over Baffin Bay is floating forming an ice shelf.

> The revised version of Figure 6 highlights the grounding line and uses color maps that are less confusing.

**Comment 22**

Figure 11c: it would be interesting to see your simulated bedrock uplift and the ice thinning for the four ice-core sites as in Figure S2 of Lecavalier et al., 2017 or Figure S15 of Tabone et al., 2024. Let's consider Camp Century for instance. From Figure 11c you simulate an Holocene uplift of ~400-500 m, so to replicate the elevation drop of ~600 m you model in Figure 4, you'd need a decrease in ice thickness more or less of the same magnitude. Is this what you simulate?

**Response:** Thank you for the comment.

Yes, indeed the simulations produce Holocene uplift at CC of 367±14 m and the ice thickness decreases by almost 1000 m.

In the revised version of the paper a new figure has been added in the appendix showing the bedrock uplift and the ice thickness for the four sites (Fig. A8).

**Comment 23**

Line 269: please correct "Smith Ice Stream" to "Smith Sound Ice Stream".

**Response:** Thank you for the comment.

Yes, that is the correct name. We will change it.

In the revised version the name has been corrected to "Smith Sound Ice Stream".

**Comment 24**

Lines 272-274: how are your isochrones compared with isochrones from Leger et al., 2023? Also, "The modeled collapse of the ice bridge in Nares Strait occurs at 4.9±0.5 ka before present,...": this is in contrast with several evidence/modeling work (e.g. Figure 15 from Leger et al, 2023, but also Lecavalier et al., 2017, England et al., 2006, )...I think this is a central point that has to be solved before we can use elevation change histories to constrain models. The timing in the bedrock uplift should reflect the timing in the deglaciation, so how can we trust the Holocene thinning curves if the modeled margin retreat lags several thousand of years the observations?

**Response:** Thank you for the comment.

That is a good question. As mentioned the timing of the retreat is late compared to England et al. (2006) and Leger et al. (2024)

> In the revised version of the paper we have included a figure that shows the modeled isocrones compared to Leger et al. (2024) (Fig. A6).

**Comment 25**

Figure 9c: please choose a discrete color palette to better show the isochrones of the last deglaciation (one color every 2kyr for example).

**Response:** Thank you for the comment.

> In the revised version of the paper we have changed the color palette to only have one color every 2kyr.

**Comment 26**

Lines 278-280: where do we see this rate of change? Also, do you mean 5000 years (and not 500 years)? At least from your video (https://av.tib.eu/media/68337) I can see by eye that the rate of change seems to follow the timing of the ocean melt scaling, which in your best simulation starts 5000 years ago.

**Response:** Thank you for the comment.

The rate of change can be seen in Fig. 8 for the ensemble member with the highest likelihood. The ensemble estimates of this rate of change can be seen in table 2. We do mean the last 500 years and not the last 5000 years.
* * *
**Comment 27**

Lines 281-282: there is a lot of uncertainty around the timing and the magnitude of the oceanic forcing, but we know (1) from evidence inferred from sediment cores, that this might have increased already several thousand of years before the Holocene Thermal Maximum (e.g. Jennings et al., 2017, Lloyd et al., 2023 ...) and (2) from paleo model simulations (e.g. Trace-21ka experiments using the NCAR-POP model), that this hasn't been uniform around Greenland during the last deglaciation. For example, warmer oceanic waters have been suggested to occur in the east/northeast of Greenlandic coasts already at the early Holocene (e.g. Lloyd et al., 2023, Werner et al., 2016, ...). In this work, the ocean thermal forcing is scaled depending on the latitude (<71°N, between and > 80°N), but this is a crude simplification. Given the importance of such a forcing in your simulations, I believe you should discuss in more detail the limitations of using a quasi-uniform oceanic forcing and the unrealistic activation of this forcing only in the last 5000 years. Specifically, describe how these factors might affect your results.
* * *
**Response:** Thank you for the comment.

Yes there is a lot of uncertainty in the oceanic forcing which is also the reason we wanted to use our rather simplified approach. The spatial dependence was taken from Aschwanden et al. (2019), while the temporal dependence was inspired by the sudden onset seen by Clark et al. (2020). Introducing a non uniform scaling to for example allow different onsets between west and east of Greenland would be interesting.

> In the revised version of the paper we have included a discussion on the reason for scaling the ocean forcing uniformly and how this might be improved (section 6.6).

**Comment 28**

Line 283: "the estimated mass loss rate shifts from a prior of -12±40 mm ka-1 to a posterior of -23±26 mm ka-1": I don't see the -12±40 mm ka-1 in Table 2. Where does this estimate come from?

**Response:** Thank you for the comment.

These estimates are not shown in table 2 but are found when excluding the simulations that utilize the temperature reconstruction from Gkinis et al. (2014), where the temperature anomaly has a peak of 4.5 K during the last 500 years.

In the revised version of the paper we have included a figure showing how temperature outliers affect our mass loss estimates (Fig. A4).

**Comment 29**

Lines 310-312: so, do you think that the assumption made by Lecavalier et al., 2017 was wrong? And if yes, could you explain better why? I think that this whole section (6.3) should be better discussed. Please follow Jessica Badgeley' comments on the uncertainty in deriving elevation changes from ice cores. I think this is a central point of your discussion: how much can we trust elevation histories only to validate model simulations if we can't quantify their uncertainties?

**Response:** Thank you for the comment.

We do not think that the assumption by Lecavalier et al. (2017) is necessarily wrong. We choose to use the surface elevation histories by Vinther et al. (2009) because these estimates are backed up by independent measurements of total gas content at CC where the surface elevation histories derived by Vinther et al. (2009) and Lecavalier et al. (2017) deviate the most.

In the revised version of the paper we have rewritten section 6.3 to include these considerations.

> **Comment 30**
>
> Line 355-358: I agree that an ice-sheet model coupled to an atmospheric one would help to investigate the response of the ice sheet to non-local climatic effects, but there are cheaper solutions that could already improve the representation of the non-uniform temperature and precipitation patterns across Greenland, e.g. the usage of a spatial variable paleo climatology (i.e. Buizert et al., 2018, Badgeley et al., 2020). See my general comments.

**Response:** Thank you for the comment.

Yes the spatially variable products of Buizert et al. (2018) and Badgeley et al. (2020) would be cheaper to implement. However, these products are based on the TraCE-21Ka climate simulations which uses the ICE-5G reconstruction for surface topography. The ICE-5G reconstruction differs from our modeled topography which is why a coupling would be interesting although computationally much more expensive.

> In the revised version of the paper we include a discussion on the use of uniform temperature anomalies (section 6.7).

> **Comment 31**
>
> Lines 366-367: I believe that "further investigations" are actually needed since paleoclimate has a primary control on the evolution of the GrIS, and potentially on the surface elevation history. See point above.

**Response:** Thank you for the comment.

We agree that as a next step it would be interesting to use a non-uniform temperature reconstruction although we believe that keeping the Holocene reconstruction of the Greenland Ice Sheet independent of any previous reconstruction would be best to avoid circular conclusions.

> In the revised version of the paper we have included a discussion on the use of uniform temperature anomalies (section 6.7).

**Comment 32**

Lines 421-427: I find this paragraph really interesting, but it should be better explored in the discussion, not only in the conclusion. For example I wasn't able to see a proper discussion on the deviation of your modeled uplift rates at the present with respect to the observed ones, besides one sentence in Section 5.4. Again, I think that the delay in the modeled retreat is a central part of the discussion and should be better addressed.

**Response:** Thank you for the comment.

We agree that this is important for the discussion. Section 6.2 includes a discussion on the discrepancies between the modeled and observed present-day bedrock uplift rates.

**Authors' Response to Referee 2**

**General Comments.** This manuscript presents simulations of the Greenland ice sheet spanning the last 20,000 years, using elevation histories derived from ice cores as constraints. 20 model parameters were systematically tested using Bayesian inference on an ensemble of 841 simulations. Aside from a glacial spin-up, all simulations use PISM at 10km resolution, which is adequate for this task. The main results is a set of model parameters that was constrained by time-varying reconstructions and the conclusion that a good fit is not possible without allowing the ice sheet to bridge over Nares Strait and connect to the Innuitian ice sheet.

I think this study is timely and highly relevant, even if the data used as constraint is not new (Vinther et al., 2009) and more comprehensive datasets exist. I come back to this point in my comments below. Overall, the manuscript presents work of high quality and the presentation, text and figures, is very good. I agree with the conclusion that long-term, transient trends in ice volume should be considered to accurately and reliably project future mass loss and sea level contribution. This work is a big step in this direction. However, similar points have previously been made MacGregor et al. (2016, doi: 10.1126/science.aab1702) who also point out the importance of the ice bridge across Nares Strait. The manuscript should reference this earlier work.

My criticism is best summarized in four major comments:

**Response:** Thank you for your feedback and interest in our work.

We will make sure to reference MacGregor et al. (2016) and their work on GrIS's multimillennial-scale response to a collapsed ice bridge across the Nares Strait.

We have carefully addressed all the issues item by item as follows.

**Comment 1**

1) Use of idealized climate

The simulations use one of five spatially uniform temperature anomalies. Precipitation is based on these anomalies and modified with a simple meridional gradient. I noted, after(!) my own reading, the extensive comments on this issue in the discussions section and will therefore keep my comment brief. However, I believe that a more in-depth discussion of this approach and its implications is needed. For example, most parameter estimates agree within their uncertainty (Table 1), but E_SIA is clearly different for the northern and southern core sites (line 212ff). I believe that this is a symptom of the model fighting systematic bias in the boundary conditions, possibly the climate.

**Response:** Thank you for the comment.

The uniform temperature anomalies and meridional gradient in the precipitation scaling is indeed an idealization of the climatic conditions. The choice of using the uniform temperature anomalies was to align with the assumption of Vinther et al. (2009) that local temperature offsets were due to a Greenland wide temperature anomaly and local elevation feedback.

We agree that the differences in the estimated model parameters could be a symptom of the model fighting systematic bias. We believe that if the model is fighting systematic biases in the climate it is reflected in the differences in the inferred forcing parameters and optimal temperature anomaly at the site.

The discrepancy between the estimates of E_SIA could also be due to the model fighting systematic biases in the supplied boundary conditions, but we find it more likely to be due to the models inability to correctly model the dynamics as mentioned in section 6.5.

> In the revised version of the paper we argue in section 6.5 that the differences in the estimated model parameters like E_SIA could possibly be due to the spatial differences in the SMB that are not included in our model.
>
> In the revised version of the paper we have included a discussion on the use of uniform temperature anomalies in section 6.6.

**Comment 2**

2) Comparison with more recent data

I think a revised version of the manuscript should include a comparison with the independent dataset by Leger et al. 2024.

**Response:** Thank you for the comment.

We think this is a very good idea as getting the timing of the retreat right is the direction in which we think our simulations could improve the most.

> In the revised version of the paper we include a figure showing a comparison between the modeled extent and the extent by Leger et al. (2024) (Fig. A6).

**Comment 3**

3) Unclear bedrock optimization

I am not sure if I understood the bedrock optimization routine correctly. Is it correct that it was only performed with one single set of parameters? The modern bedrock topography of the best fit simulation shows a substantially larger deviation from observations (Fig. 11b) and a higher RMSE than the simulation used to initialize the ensemble. Why is this the case and what implications does that have?

**Response:** Thank you for the comment.

Yes, the bedrock optimization is only done for one set of parameters and at 20 km resolution. It was done because the deviation between the modeled present-day bed topography deviated substantially from the observed one. The modeled bedrock topography deviates from the observed because the parameters are changed and the resolution increased. Ideally the bedrock optimization should be done for each member of the ensemble but this would computationally much more demanding. Updating the bedrock does effect the modeled elevation histories.

> In the revised version of the paper we have clarify the bedrock optimization routine described in section 3.5 and included a figure showing the effect of updating the bedrock topography (Fig. A7).

**Comment 4**

4) Conclusions on ice bridge

The two simulations that restrict the GrIS from advancing beyond the present-day coastline or the ECS mask were run with the same parameters as the best fit simulation from the ensemble without any spatial restrictions (line 202f). Can strong conclusions be drawn from such a setup? How can you exclude the possibility that a different climate is compatible with the ice core constraints without the need to limit the ice extent?

**Response:** Thank you for the comment.

We cannot exclude the possibility that a different climate forcing and different model parameters would be compatible with the surface elevation histories. However, two simulations serve to show the effect of not allowing the ice sheet to advance beyond the present-day land mask or the continental shelf. Neither simulation has an ice sheet that is thick enough at Camp Century before the onset of the Holocene. While this might be remedied by decreasing the viscosity or increasing the precipitation, this would be modeling the correct elevation for the wrong reason as the geological evidence shows that the ice bridge existed. Thus, our simulations propose that the ice bridge formed.

> In the revised version of the paper we have included this consideration in the discussion (section 6.1).

Minor comments:

**Comment 5**

equations 1 and 3: It is not clear how the latitudes were chosen and why they are different.

**Response:** Thank you for the comment.

The latitudes in equation 1 were chosen to cover most of Greenland, while the latitudes of equation 3 were chosen to be the same as those used by Aschwanden et al. (2019) who used this parametrization based on observations along the western coast of Greenland.

> In the revised version of the paper we have clarified why the choices of latitudes were made.

**Comment 6**

l 140: Please include an explanation why E_SIA and n_SSA are varied but not E_SSA and n_SIA.

**Response:** Thank you for the comment.

We follow Aschwanden et al. (2022) in the choice of varying E_SIA and n_SSA and not E_SSA and n_SIA as PISM is optimized for n_SIA=3.

In the revised version of the paper we have clarified why we vary E_SIA and n_SSA and not E_SSA and n_SIA.

**Comment 7**

l 221: I think this should read "The northern 'precipitation' parameter, not 'accumulation'.

**Response:** Thank you for the comment.

In the revised version of the paper we have changed all instances of 'accumulation parameter' to 'precipitation parameter'

**Comment 8**

l 289f: "[..] modeled bedrock topography is very sensitive to the history of the ice load." Please see my comment #3 above.

**Response:** Thank you for the comment.

In the revised version of the paper we have clarified the bedrock optimization routine described in section 3.5 and included a figure showing the effect of updating the bedrock topography (Fig. A7).

**Comment 9**

section 3 could be more explicit about the model resolution.

**Response:** Thank you for the comment.

In the revised version of the paper we have made sure to state the model resolutions used in the beginning of section 3.

**Comment 10**

section 6.6 needs to discuss the implications of using a uniform temperature anomaly forcing, maybe after presenting the findings by Lecavalier et al. (2017) (line 362). Please consider the comment by Jessica Badgeley.

**Response:** Thank you for the comment.

In the revised version of the paper we will include a discussion on the use of uniform temperature anomalies (section 6.7).

**Comment 11**

l 355f: I think it is too strong to state that a full coupling of ice sheets to atmosphere and ocean is the only alternative to including spatially non-uniform climate forcing.

**Response:** Thank you for the comment.

That is correct. There are cheaper options available as also made clear by the other referee.

In the revised version of the paper we have changed the wording of line 355 and mention the other choices in section 6.7.

**Comment 12**

figure 5 (and others): The line colors, in particular the two shades of blue, are difficult to distinguish.

**Response:** Thank you for the comment.

The colors used in our figures are taken from the color palette described in Wong, B., 'Points of View: Color Blindness,' Nature Methods 8, 441 (2011). https://doi.org/10.1038/nmeth.1618. This palette offers good variability in lightness, saturation, and hue, making the colors easily distinguishable by individuals with red-green color blindness.

In the revised version of the paper we have changed the sky blue to orange in Fig. 5

**Authors' Response to Community comment 1**

**General Comments.** Dear authors,

I just wanted to point out a potential minor error in the use of the Leger et al. (2024) shapefile data.

In the preprint figure 9 panel C: the red outline is described by the authors as the "LGM extent from Leger et al. (2024)": whilst it looks more to be the outermost isochrone we mapped: the 14-13 kyr BP isochrone. This isochrone represents a less extensive GrIS than the full extent of the GrIS reached during the LGM, which occurred a few thousands years before (between 18 and 15 kyr BP): and which you can see mapped in figure 5 panel B of Leger et al. (2024). In the latter figure one can see we propose two scenarios for the full LGM GrIS extent from data: due to remaining uncertainties in certain regions. Regardless of which scenario you choose (min or max), this LGM extent will most likely fit your modelled LGM better: so I would advise re-making the comparison with this LGM extent rather than the 14-13 kyr BP isochrone, which is not quite the LGM.

The shapefile for the LGM extent can be found in the PaleoGrIS database under :

PaleoGrIS_1.0_isochrones\Shapefile_format\Full_Glacial_max_min_literature_review

Let me know if I've missed something and am mistaken,

Best wishes, and congrats on the work and paper which I will follow with much interest.

Tancrede Leger

**Response:**

Dear Leger

Thanks for pointing this out and thank you for your interest in our paper. I have now updated the figure to include both your minimum and maximum LGM extent. Our modeled LGM extent is 0.9% larger than your minimum and 5.6% smaller than your maximum.

**Authors' Response to Community comment 4**

> **General Comments.** This study contributes to the ongoing and necessary efforts to use more data, and a greater diversity of data, to inform ice sheet models. Data constraints are sparse far from the margins of ice sheets. It is therefore especially important to leverage interior data, such as ice core measurements, as has been done in this study. That said, there are many challenges in doing this, some of which I think could be addressed or discussed more thoroughly in this manuscript.
>
> I would be happy to discuss any of this in further detail. Thanks again for your contribution to this rapidly evolving field.
>
> Jessica Badgeley

**Response:**

Dear Jessica Badgeley,

Thank you for your comments and interest in our paper.

We have carefully addressed all the issues item by item as follows.

> ### Comment 1
>
> Lines 53-59 and 305-320: Estimating elevation from paleotemperature records is sensitive to the choice of lapse rate and any atmospheric circulation changes that occur at the same time as the elevation change (e.g., Forest, 1995; Meyer, 2007; Badgeley et al., 2022). Vinther et al. (2009) discuss this, but my understanding is that neither of these known uncertainties are quantified or included in their estimate of the elevation history uncertainty. If this is correct, are these additional uncertainties quantifiable? How might they impact the findings of this study?

**Response:** Thank you for the comment.

The uncertainty of the surface elevation histories by Vinther et al. (2009) propagates from the spread of O-18 values of two parallel records from the Agassiz ice cap and the uncertainty in bedrock uplift at the Agassiz and Renland ice cap locations. The surface elevation estimates of Vinther et al. (2009) do indeed use a constant lapse rate of -0.6‰ per 100 m which directly translate the O-18 signal into an elevation signal. The uncertainty in this value is not estimated and introducing an uncertainty would increase the spread of the surface elevation histories and in turn our inferred probability density functions.

In the revised version of the paper we have included these considerations in our discussion in line 178 after introducing the likelihood function.

**Comment 2**

Lines 60-70: Do the temperature histories from Nielsen et al. (2018) subtract out the impact of elevation change? If so, it would be helpful to state this. If not, then there appear to be signals in the d18O/temperature records that are being double counted in this study because the Nielsen et al. (2018) temperature anomalies are applied at a constant (modern) elevation. By "double counting" I mean that, in the ice core data, a part of the d18O signals is attributed to elevation change, while in the SMB forcing this same part is attributed to temperature change at a constant elevation.

**Response:** Thank you for the comment.

Only the temperature reconstruction of Vinther et al. (2009) subtracts the impact of elevation change, but the other reconstructions were calibrated to measured bore hole temperatures or to the measured isotope diffusion record, thereby avoiding assumptions of the isotope to temperature relation. We choose to also sample over the other reconstructions used by Nielsen et al. (2018) to cover the historical temperature uncertainty, despite the limitations in these records.

In the revised version of the paper we state that only the temperature reconstruction by Vinther et al. (2009) is corrected for elevation changes in lines 60-70 when introducing the temperature anomalies.

**Comment 3**

Lines 114-121: There are ice-core informed Greenland paleoclimate reconstructions that provide spatially variable histories of temperature and precipitation (e.g., TraCE-21ka – Liu et al., 2009 and He et al., 2013; Buizert et al., 2018; Badgeley et al., 2020). These reconstructions make it unnecessary to apply a constant temperature scaling or to scale precipitation from temperature. Though these studies show different spatial patterns of temperature change over Greenland, they all show that there is not a single, spatially constant temperature scaling.

**Response:** Thank you for the comment.

The choice of using the uniform temperature anomalies was to align with the assumption of Vinther et al. (2009) that local temperature offsets were due to a Greenland wide temperature anomaly and local elevation feedback.

The TraCE-21K climate simulations use the ice sheet configuration of ICE-5G which is not consistent with our simulated ice sheet and does not include an ice shelf covering Baffin Bay. Placing too high confidence in these products introduces a range of additional uncertainties and assumptions. When making a reconstruction of the Holocene evolution, we prefer to keep our method simple, dependent of as few parameters as possible, and independent of any previous reconstructions to avoid any circular conclusions.

> In the revised version of the paper we include a discussion on the use of uniform temperature anomalies (section 6.7).

**Comment 4**

Lines 178-179: I think this statement needs more explanation or justification. Why not assign a lower uncertainty to present-day data since it is much more certain than the paleo constraints? Separately, why not include other modern and paleo data, such as satellite data and exposure ages of moraines? These datasets have been used before (e.g., Briggs and Tarasov, 2014; Briner et al., 2020) and would provide greater constraints on the model parameters.

**Response:** Thank you for the comment.

Including present-day observables would improve confidence in the present-day state. However, our focus is on testing the feasibility of using the past surface elevation history to understand how prognostic (future) simulations might be biased by the long-term response of the GrIS to past climatic changes. To judge this, we argue that using this data in isolation makes our results most clear. We therefore also choose to assign equal uncertainty to all points in time as we are interested in modeling the transient response to the past climatic changes and our aim is not to capture the exact present-day state.

Including the exposure ages of the moraines would indeed increase our confidence in this transient response. However, our simulations are systematically late in modeling the lateral retreat. Combining the interior surface elevation records with the exposure ages of the moraines would be a next step.

> In the revised version of the paper we clarify why we only use the surface elevation histories as constraints in our Bayesian inference and why we assign equal uncertainty to all points in time (section 4).

**Comment 5**

Lines 202-206: The conclusions drawn from the "restricted to GrIS" and "restricted to ECS" models may be correct, but I believe the comparison of these simulations to the others is unfair. The best-fit parameters for the main model will not necessarily be the best for either of the restricted models. If running a separate parameter calibration for each restricted model is beyond the scope of this study, then, at a minimum, it would be useful to use a smaller ensemble to determine whether the RMSEs for the different parameter combinations correlate across the three models. If they do, then perhaps the conclusions stated in lines 202-206 are justified by the current set of simulations.

**Response:** Thank you for the comment.

We don't claim that the best-fit parameters for the main model will be the best for either of the restricted models. The point is simply to show the effect of allowing the ice sheet to advance beyond the present-day land mask and form an ice bridge to the IIS. Neither of the restricted simulations has an ice sheet that is thick enough at Camp Century before the onset of the Holocene. While this might be remedied by decreasing the viscosity or increasing the

precipitation, this would be modeling the correct elevation for the wrong reason as the geological evidence shows that the ice bridge existed.

**Comment 6**

Figure A2: I find the color scheme for this figure to be counterintuitive. Standard practice in my experience is that colder temperatures are shown in blue, warmer temperatures in red, and greater precipitation rates in darker blues.

**Response:** Thank you for the comment.

We agree that the color schemes can be counterintuitive and will change these.

In the revised version of the paper the colormaps in Figure A2 are changed.

**Authors' Response to Community comment 5**

**General Comments.** The authors do a great effort to constrain the Holocene history of the Greenland Ice Sheet, a timely and valuable effort. Unfortunately, with respect to the NE sector of the GrIS, they do not include references to other recent and highly relevant findings derived from airborne radar observations for this topic, e.g. Franke et al (https://www.nature.com/articles/s41561-022-01082-2) or Jansen et al (https://www.nature.com/articles/s41467-024-45021-8). In fact, those analyses would greatly profit from this manuscript to revisit the provided interpretations. At the same time, this manuscript could provide a more precise interpretation than currently presented. More spedifically, a sentence like "During the Holocene collapse of the IIS, the ice divide at the GrIS moves towards the west and the ice streams reorganize in northern Greenland as shown in Fig. 6" would profit from the mentioned references, where divide migration has already been postulated as a potential mechanism.

Thanks for consideration

Olaf Eisen

**Response:**

Dear Olaf Eisen,

Thank you for pointing out the works by Franke et al. (2022) and Jansen et al. (2024). In the revised version of the paper we point out how our ice divide migration and rearrangement of ice streams align with the shutdown of a paleo NEGIS and a turnon of the present-day NEGIS.

---

## Author Response (AR2)

**Modeled Greenland Ice Sheet evolution constrained by ice-core-derived Holocene elevation histories**

Addressed Comments for Publication to

**The Cryosphere**

by

Mikkel L. Lauritzen, Anne Solgaard, Nicholas Rathmann, Bo M. Vinther, Aslak Grindsted, Brice Noël, Guðfinna Aðalgeirsdóttir, Christine S. Hvidberg

**Authors' Response to Referee 1**

> **General Comments.** Thank you for submitting your revised manuscript. You added content to address the comments of the reviewers. However, I still find many places where the manuscript can be significantly improved. Moreover, there is still an important point regarding the explanation of the elevation reduction at CC that is not convincingly treated. After reading your revised manuscript, it became clear that this is more problematic than I recognized in my initial assessment and should be treated carefully. Below I list this, a few other important points, and then several more specific comments. Please address these comments in a revised submission.

**Response:** Thank you for your feedback and interest in our work.

We have carefully addressed all the issues item by item as follows.

**Comment 1**

**Missing explanation for ice-thickness reduction**

You make the argument that the focus of this work is to reproduce the elevation histories at the ice-core locations, and therefore the poor margin-retreat timing is not relevant here. This is potentially a reasonable idea, but in practice, it is not very credible given that you specifically relate a large part of the elevation change at CC to growth of the ice bridge across Nares Strait. For the growth of the ice bridge, this is convincingly treated, and the tests with restricted ice sheets support this argument nicely.

For the retreat, however, you treat this sparingly (L393-396), and this needs much more clarification. It is, at first, wholly unclear how it is possible obtain such good agreement in timing with the elevation reduction at CC, when the ice bridge does not collapse until after -7 ka. One would expect the collapse of the ice bridge to drive the elevation reduction. Nor does "reducing buttressing" make sense since the large ice shelf in Baffin Bay remains until about -7 ka too. This point was noted by Reviewer 1 too.

But I think the SI movie shows what is happening and I believe this deserves much more attention in the results and discussion, because it is very interesting. Namely, just before -9 ka, a very large paleo ice stream forms inland from Baffin Bay and accelerates ice flow all along the northwest coast (this can also be seen comparing Fig. 6a and 6b). It would appear this is the primary reason you can match the elevation reduction timing, and it is quite novel. Furthermore it links nicely with the recent work of Tabone et al (2024) - disclaimer I am a coauthor - who link elevation change to an acceleration of the paleo NEGIS.

Therefore I would propose you analyze and describe the formation of this ice stream and its branches. What triggers its growth? Is ice becoming temperate at the base as it gets thicker and insulated, while the climate also warms? Or, is ice just getting thinner until causing widespread grounding-line retreat, accelerating flow? Did the southern branch of the ice-stream also affect NGRIP (complementing/contrasting with the results of Tabone et al.)? Could there be evidence for such an extensive ice stream? It seems that such an ice stream could only form with the large growth of Nares strait too, so it supports the general message of the paper - that simulating this feature is important.

Perhaps I am wrong, and there is another explanation, but then you still need to convincingly explain how the rapid elevation reduction occurs.

**Response:** Thank you for the comment.

Thank you for your comments on this, in particular the remarks about the paleo–ice stream in Baffin Bay and its effect on Camp Century thinning. We agree that this provides a convincing explanation for the timing of thinning at Camp Century, and we appreciate you pointing this out.

To accommodate this, we have revised the order of subsections in the Discussion and rewritten the first two subsections. The first subsection now provides a consistent explanation for the two stages of interior thinning, attributed to the formation of the paleo–ice stream and the subsequent collapse of the ice bridge, respectively. We clarify where our model results agree with other data and where they differ, and we explain why our model can reproduce the timing of thinning at Camp Century, even though the retreat timing at the margin differs. We describe how the paleo-ice stream forms as surface melting sets in and the Baffin Bay ice shelf breaks up. We further describe how it is linked to the bedrock topography along the coast. We have also added a reference to Tabone et al. (2024). Overall, we find that this explanation is now significantly improved.

The second subsection has also been revised. It now presents a more detailed discussion of the Holocene evolution of ice volume and spatial patterns, and concludes with a brief discussion of the importance of paleo calibration for constraining modern mass loss rates.
* * *
**Comment 2**

The section "2 Paleoclimatic evidence" does not really belong as it is now. The title is not really clear - evidence of what? The elevation histories were introduced earlier, as were temperature reconstructions. I would propose the following: - Move the content of the first paragraph L58-64 into the Introduction, probably around L22 when "temperature reconstructions vary by several degrees" is stated. - Make a new subsection within the section "3 Model setup", which would be "3.1 Atmospheric forcing". Move the second paragraph there to describe the temperature forcing used. Additionally move the temperature information and precipitation scaling information from "Surface mass balance" to "Atmospheric forcing", for more consistency.

**Response:** Thank you for the comment.

We agree that it is better to divide Section 2 into the Introduction and the Model Setup sections.

We have incorporated lines 58–64 into the part of the Introduction that introduces the elevation histories. The part describing the temperature forcing has been merged with the "Surface mass balance" section which is now titled "Atmospheric forcing" for improved consistency as suggested.

**Comment 3**

I acknowledge the simplicity of the model setup here and I appreciate it. However, I think we are now confident that the PDD model lacks realism when insolation changes significantly (see e.g. Robinson and Goelzer, 2014). It is clear that during the early Holocene particularly at high-latitude sites like CC, the additional contribution to melting from increased insolation would be important. There are now multiple approaches available to incorporate insolation forcing, including the dEBM-simple model that is part of the PISM package. You should acknowledge this deficit in the "Surface mass balance" section, and include a paragraph about the possible implications of accounting for insolation changes on melting in the Discussion.

**Response:** Thank you for the comment.

We agree that the PDD model definitely lacks realism, particularly when insolation changes significantly.

We have acknowledged this in the new section, "Atmospheric Forcing" and include a paragraph about the insolation changes in the "Climatic forcing" section of the discussion.

**Comment 4**

Note: a hyphen should generally be used when a compound noun becomes and adjective, so "ice sheet" versus "ice-sheet model". I noted several instances below, but please try to check the manuscript throughout.

L2: Ice sheet reconstructions → Ice-sheet reconstructions

L12: ice sheet evolution → ice-sheet evolution

L19: an abrupt warming → abrupt warming

L28: ice sheet model studies → ice-sheet model studies

L32: ice sheet model → ice-sheet model

L34: long term response → long-term response

L39: Before the satellite era → To simulate time periods before the satellite era,

L39: ice sheet modeling → ice-sheet modeling [further instances not noted here]

L43: and significantly → and could significantly

L51: key model parameters → influential model parameters

**Response:** Thank you for the comment.

We have tried our best to hyphenate all the compound nouns and incorporated all the proposed changes.

**Comment 5**

L17: Please replace reference to Gulev et al., 2021 with direct reference(s) to support the given range.

**Response:** Thank you for the comment.

We have replaced the IPCC reference with Lambeck et al. (2014) and Yokoyama et al. (2018)

**Comment 6**

L22: "which can significantly affect the modeled GrIS" ← You have not yet introduced modeling. Please try to rephrase here.

**Response:** Thank you for the comment.

> This line has been rephrased.

**Comment 7**

L109: "We use two constants of proportionality" ← These should be accompanied by an equation, otherwise it is not clear what they refer to. As they are parameters tested in the ensemble, this is especially relevant.

**Response:** Thank you for the comment.

> We have added equation for the surface melt in the new section "Atmospheric Forcing".

**Comment 8**

L113: "We looked at three areas for rainfall data" ← This sentence is not clear to me at all. I understand both temperature and precipitation are obtained from the RACMO simulations. Why is "rainfall data" needed in specific regions? Furthermore, should this be precipitation, or specifically rainfall? If the latter, what is its relevance? Please make this paragraph more clear on these points.

**Response:** Thank you for the comment.

This is indeed a blunder. It should be precipitation and not rainfall.

> The sentence has been rephrased

> **Comment 9**
>
> L121: Explicitly define $\omega_\uparrow$ and $\omega_\downarrow$ as free parameters in the text, since these parameters are also modified in the ensemble.

**Response:** Thank you for the comment.

> We have defined $\omega_\uparrow$ and $\omega_\downarrow$ as free parameters in the text.

> **Comment 10**
>
> L131, Eq. 3: Rather confusing that $\phi_\uparrow$ and $\phi \downarrow$ have the same name as in Eq. 1, but different values. Add a subscript "o" or similar to distinguish.

**Response:** Thank you for the comment.

We agree that this was confusing.

> We have added superscripts "o" for ocean and "p" for precipitation. We have also added the "o" superscript to the ocean melt rate to distinguish it from the surface melt rate $\dot{m}^s$ We also changed the friction angle $\phi$ to $\varphi$ to distinguish it from the latitude.

> **Comment 11**
>
> L145: I see that n_sia is fixed to 3, while n_ssa is modified. Changing n_ssa is not so easily done, however, without also adjusting the constant factor A (since its units must be proportional to n). See e.g. Zeitz et al. (2022), who do this in a consistent way using PISM. Is this the method that is used? If so, it should be cited. While it is true that Aschwanden and Brinkerhoff (2022), who you cite, also state that they vary this parameter, it is not explained there either how this is achieved, while maintaining consistency.

**Response:** Thank you for the comment.

Yes, n_SIA is fixed to 3 while n_SSA is modified. The numerical value of A is not changed, only the units are adjusted, which indeed modifies the viscosity for effective stresses different

from 1 Pa. For an effective stress of 80 kPa, a change in n by 0.2 reduces the viscosity by approximately a factor of 10, corresponding to an enhancement factor of 10. We consider this not an inconsistency, but rather a deliberate modeling choice.

We have clarified that A is not varied.

**Comment 12**

Figure 2: I like this figure as a schematic to show what you did. But I think it could be improved if the red dashed line would point rather to the lower -20 ka dot, to show that this is an iterative loop. Then you could have a black dashed line going from PD to the upper -20 kyr dot, which indicates it is not part of the adjustment process, but rather now you move forward with your ensemble of simulations. This is only a suggestion, of course.

**Response:** Thank you for the suggestions

We have changed Fig. 2 according to your suggestions.

**Comment 13**

Figure 2, caption: the last glacial → the last glacial period

**Response:** Thank you for the comment.

This has been changed.

**Comment 14**

L153: Please rephrase slightly. The entire equation has been simplified, not just the reference pressure, as the additional term also included a dependence on W_till/W_till_max. Was there a particular reason to simplify the equation in this way?

**Response:** Thank you for the comment.

The original formulation of Bueler and van Pelt (2015) the effective reference pressure before being capped at $P_0$ is

$$\hat{N}_{\text{till}} = N_0 \left( \frac{\delta P_0}{N_0} \right)^s 10^{\left( \frac{e_0}{C_c} \right)(1-s)},\tag{1}$$

which can be written as

$$\hat{N}_{\text{till}} = 10^{\left( \frac{e_0}{C_c} \right)} N_0 \left( \frac{\delta P_0}{10^{\left( \frac{e_0}{C_c} \right)} N_0} \right)^s\tag{2}$$

$$= \tilde{N}_0 \left( \frac{\delta P_0}{\tilde{N}_0} \right)^s,\tag{3}$$

where $\tilde{N}_0 = 10^{\left( \frac{e_0}{C_c} \right)} N_0$. This is what the effective pressure at zero void ratio or saturation would be if it was not capped at $P_0$. $e_0$ is the void ratio at the reference pressure $N_0$ and $C_c$ is the till compressibility. I think that using $\tilde{N}_0$ makes it easier to read and the interested reader can then consult the literature.

> We have rephrased the part of the manuscript describing the effective pressure to make it more readable

**Comment 15**

L157: Please justify these bedrock choices somewhat. Why are neither of these parameters considered in the ensemble? Arguably, they could have an impact on the transient evolution of the ice sheet's elevation history.

**Response:** Thank you for the comment.

Arguably we could have varied the viscosity and lithosphere flexural rigidity, we would then have had to adjust the bedrock individually for all ensemble members further adding to the computational requirements of the study. We mention this in the discussion section "Bedrock uplift".

**Comment 16**

L162: "Since we only know the bedrock elevation for the present day and not for the past" ← I think the application of this approach requires a bit more justification than this, especially given that you run Greenland transiently through the glacial cycle beforehand. You could arguably say the same thing about any number of the model choices made, but in some cases you include the parameters in the ensemble, whereas here the bedrock parameters are fixed. Why in particular should the errors associated with the bedrock approach be artificially reduced? Has this or a similar approach been used before? If so please include citation.

**Response:** Thank you for the comment.

You are right that the errors are indeed artificially reduced, and this should be better justified.

The reason for this approach is that the modeled ice extent, and consequently the surface elevation, is sensitive to ocean melt and sea level forcing. We apply an artificial correction to the bed topography to ensure that the present-day ocean mask aligns closely with observations. This correction improves our ability to reproduce the elevation change at Dye 3, which we were initially unable to match without it, as shown in Fig. A6.

Our approach is similar to the scheme used by van Calcar et al. (2023), whom we will cite.

> We have changed this section to better justify the need to artificially reduce the bedrock elevation error and to refer to van Calcar et al. (2023).

**Comment 17**

Figure 3: Panel e still shows the 3rd iteration bedrock difference, but I understood this would be replaced with that of the last iteration. Was this an oversight, or a different decision?

**Response:** Thank you for the comment.

This was indeed an oversight, we forgot to include it.

> Panel e of Fig. 3 now shows the last iteration.

**Comment 18**

L172: surface elevation → bedrock elevation

**Response:** Thank you for the comment.

This was correct. Fig. A7 shows the effect on the surface elevation to the bedrock changes

**Comment 19**

L174: 20 key parameters → 20 parameters

**Response:** Thank you for the comment.

We have changed this.

**Comment 20**

L184, Eq. 9: I think it would make more sense to introduce Eq. 11 first, but it is just a suggestion.

**Response:** Thank you for the comment.

We prefer to keep it this way.

**Comment 21**

L203, Eq. 12: Is this a standard approach? Perhaps define what it means in words. Also, I note the only place it appears later is at the end of Table 2 without any clarification.

**Response:** Thank you for the comment.

This is a standard approach for evaluating the efficiency of the sampling. The effective sample size is the number of equally weighted samples that would yield the same variance of the mean as the weighted set of samples. If only one ensemble member has a non-zero likelihood, the effective sample size is one. Conversely, if all members have the same likelihood, the effective sample size equals the actual sample size, namely 841.

We mention the effective sampling size in the discussion "Sampling technique".

We have tried to explain the meaning of the effective sampling size better. The subsection "Sampling technique" has been merged with "Inferred parameters"

**Comment 22**

L210-214: It is not clear to me from the labels what the difference is between the modeled "individual estimates" and the "combined estimated elevation". The elevations shown are specific to each site. So, does "individual estimates" refer to envelope of simulations that best match the elevation change at that specific ice core (while any mismatch with other ice cores is not accounted for), and the "combined estimated elevation" is then the envelope of simulations that best match the elevation changes of all cores together? If so, please try to rephrase here and in figures. Perhaps this could be something like "Site-specific pdf" and "Combined pdf", or even "Site-specific estimate" and "Combined estimate". Try to make the language precise and consistent between the two cases. And in place of "Ice-core-derived", I would simply put "Reconstruction".

**Response:** Thank you for the comment.

We agree that "site-specific pdf" and "Combined pdf" is clearer. We prefer however to keep the term "Ice-core-derived" to avoid confusing it with our "reconstruction"

We have changed the labels in the figure and rephrased the text.

**Comment 23**

L222: RMSEs to → RMSEs associated with

**Response:** Thank you for the comment.

This has been changed.

**Comment 24**

L228: "Notably, the estimated enhancement factor of the SIA, E_SIA, differs substantially between the sites." ← Try to improve the wording of sentences like this one, which is currently ambigious. Right now, it sounds like E_SIA is varying spatially with a different value at each ice-core site. I think what is meant is that the most likely enhancement factor value changes substantially depending on which site's elevation history is used as a target. This happens elsewhere, for example, in the very next two sentences too. Please check throughout.

**Response:** Thank you for the comment.

We agree that the wording was ambiguous.

> We have changed the phrasing so it is clear that it is the site-specific estimates that are different and not the parameters that are varying spatially.

**Comment 25**

L272: Solgaard and Kusk (2023) ← Cite the paper where these data are published: https://essd.copernicus.org/articles/13/3491/2021/

**Response:** Thank you for the comment.

> This has been changed.

**Comment 26**

L272-280: Please revise these paragraphs to ensure the figures support the text, as opposed to the text supporting the figures.

**Response:** Thank you for the comment.

> We have revised the paragraphs to ensure that the figures support the text and not the other way around.

Furthermore, we have rearranged the figure panels such that the velocity difference, previously shown in Figure 10c, now appears in Figure 9c. Panels 10a and 10b have been removed, as they did not contribute meaningfully to the text. The isochrones that were previously in Figure 9c are now shown in Figure 10, alongside the isochrones from Leger et al. (2024).

**Comment 27**

Figure 6: Nice figure. It would be instructive to add a panel here showing the distribution at LGM as well, despite the focus of the paper being the Holocene. It is a reference point of interest that is discussed in the text. If you believe it doesn't fit here, then in the Appendix, perhaps together with Fig. A9, which is explicitly related to the LGM. Also, it would make sense to show the dot locations of the ice cores too.

**Response:** Thank you for the comment.

We have added the ice-core sites to Fig. 6.

We have added the ice sheet configuration at -20 ka to Fig. A9 which is already showing the configuration at -20 ka. This figure will then show the state at the branch-off point.

**Comment 28**

Figure 7: Please add points that correspond to the estimate of Leger et al. (2024) - see their Figure 16. An explicit discussion of this mismatch with your own should be added (see general comment above).

**Response:** Thank you for the comment.

Good idea.

We have added the points from Leger et al (2024) to Fig. 7

The discrepancy is now discussed in section 5.1

**Comment 29**

Fig. 9b: I think the ice deviations are incorrectly limited to the present-day border of the ice sheet. If there is a good reason to do so, this should be clarified in the caption.

**Response:** Thank you for the comment.

That is correct. There was no good reason for this.

> We have plotted the thickness deviation for the entire domain.

**Comment 30**

L306: "historical calibration" ← Do you mean paleo calibration? Historical typically refers to the current period with direct observations.

**Response:** Thank you for the comment.

You are right.

> We have changed the instances of historical to paleo where it made better sense.

**Comment 31**

L335: "Lecavalier et al. (2017) presented revised temperature anomalies for the Agassiz ice core that, in turn, increased the ice-core-derived surface elevation at CC by 400 m at the Holocene onset." ← Please then link this to your results. Is the Lecavalier et al. (2017) estimate not plausible and therefore not considered? Or it is plausible, as well as Vinther et al. (2009), and would imply X?

**Response:** Thank you for the comment.

The Lecavalier et al. (2017) estimate is also plausible and would imply that our model underestimates the elevation which might be remedied by increased precipitation.

> We have revised this subsection to link Lecavalier's work to our results.

**Comment 32**

L337-340: It seems to me that this paragraph should be combined with the first one that also discusses the Vinther et al. (2009) methods and assumptions, and also O-18.

**Response:** Thank you for the comment.

The paragraphs have been combined.

**Comment 33**

L339: "The modeled O-18 should then be compared directly with the observed values at the time-dependent ice core site location." ← Same comment. Can you link this back to your work? E.g., Our work rests on the assumption that any potential changes in moisture sources did not materially impact the O-18 fractionation over the ice sheet. [And if you combine this as suggested with the first paragraph, you can still conclude with the comment about total gas content, which would seem to support this assumption.]

**Response:** Thank you for the comment.

This has been combined with the first paragraph linking it to our work and concluding with the comment on total gas content.

**Comment 34**

L382: "Once this issue is addressed" ← What issue are you referring to here? The late retreat of your simulations? Please be more explicit.

**Response:** Thank you for the comment.

It is the issue of having to much thinning at CC for ensemble members with earlier onset of ocean forcing.

We have revised the sentence to be more specific.

**Comment 35**

L446: Again, is "historical" meant here or "paleo"?

**Response:** Thank you for the comment.

> We have changed this to "paleo".

**Comment 36**

L455: dynamical response → dynamic response

**Response:** Thank you for the comment.

> We have changed this.

**Comment 37**

Figure A9: It would be valuable to add a panel which shows the anomaly in bedrock elevation w.r.t. present day to be able to understand more. Furthermore, I would even suggest to add a panel of the anomaly between this LGM bedrock elevation and that of a fully equilibrated bedrock to the LGM ice sheet load. This could be helpful in the discussion of your iterative approach and show how much you estimate the LGM bedrock was in disequilibrium at LGM.

**Response:** Thank you for the comment.

The modeled bedrock at -20 ka is very close to steady state. We have added a panel showing the uplift rates at -20 ka to show this.

> We have added a panel showing the anomaly from 20 ka to present day and a panel showing the uplift rates in what is now Fig. A8.

**References**

Bueler, E. and W. van Pelt (June 2015). "Mass-Conserving Subglacial Hydrology in the Parallel Ice Sheet Model Version 0.6". In: *Geoscientific Model Development* 8.6, pp. 1613–1635. DOI: 10.5194/gmd-8-1613-2015 (cit. on p. 10).

Lambeck, Kurt, Hélène Rouby, Anthony Purcell, Yiying Sun, and Malcolm Sambridge (Oct. 2014). "Sea Level and Global Ice Volumes from the Last Glacial Maximum to the Holocene". In: *Proceedings of the National Academy of Sciences* 111.43, pp. 15296–15303. DOI: 10.1073/pnas.1411762111 (cit. on p. 6).

Tabone, Ilaria, Alexander Robinson, Marisa Montoya, and Jorge Alvarez-Solas (July 2024). "Holocene Thinning in Central Greenland Controlled by the Northeast Greenland Ice Stream". In: *Nature Communications* 15.1, p. 6434. DOI: 10.1038/s41467-024-50772-5 (cit. on p. 4).

van Calcar, Caroline J., Roderik S. W. van de Wal, Bas Blank, Bas de Boer, and Wouter van der Wal (Sept. 2023). "Simulation of a Fully Coupled 3D Glacial Isostatic Adjustment – Ice Sheet Model for the Antarctic Ice Sheet over a Glacial Cycle". In: *Geoscientific Model Development* 16.18, pp. 5473–5492. DOI: 10.5194/gmd-16-5473-2023 (cit. on p. 11).

Yokoyama, Yusuke, Tezer M. Esat, William G. Thompson, Alexander L. Thomas, Jody M. Webster, Yosuke Miyairi, Chikako Sawada, Takahiro Aze, Hiroyuki Matsuzaki, Jun'ichi Okuno, Stewart Fallon, Juan-Carlos Braga, Marc Humblet, Yasufumi Iryu, Donald C. Potts, Kazuhiko Fujita, Atsushi Suzuki, and Hironobu Kan (July 2018). "Rapid Glaciation and a Two-Step Sea Level Plunge into the Last Glacial Maximum". In: *Nature* 559.7715, pp. 603–607. DOI: 10.1038/s41586-018-0335-4 (cit. on p. 6).

---

## Author Response (AR3)

**Modeled Greenland Ice Sheet evolution constrained by ice-core-derived Holocene elevation histories**

Addressed Comments for Publication to

**The Cryosphere**

by

Mikkel L. Lauritzen, Anne Solgaard, Nicholas Rathmann, Bo M. Vinther, Aslak Grindsted, Brice Noël, Guðfinna Aðalgeirsdóttir, Christine S. Hvidberg

**Authors' Response to the Editor**

> **General Comments.** You have thoroughly addressed the last round of comments, and I believe the manuscript is near ready for publication. Please find an additional set of minor comments below, which should be addressed.

**Response:** Thank you for your feedback and interest in our work.

We have carefully addressed all the issues as follows.

**Comment 1**

L11: "that the ice bridge collapsed 4.9±0.5 ka ago and" ← I think you should delete this phrase from the abstract. It is true that your simulations show this, but you also show that the timing is much too late. Highlighting it here as a confirmed result is misleading.

**Response:** Thank you for the comment.

> This has been removed.

**Comment 2**

Fig. 1: Consider including your extended glacier catchment basins as a supplement data file and script to generate it. It could be a very useful resource for other modelers in the future.

**Response:** Thank you for the comment.

Yes good idea.

> We have added a link to an archive (https://doi.org/10.5281/zenodo.15681862) under "Code and data availability"

**Comment 3**

L69: grounding line advance → grounding-line advance

L70: Model parameters, listed in Table 1, will be → The model parameters, listed in Table 1, are

L77: mass loss rates → mass-loss rates [and throughout]

L92: surface melt rate → the surface melt rate

L119: will be varied → are varied [modify to present tense throughout ensemble description]

L128: we will vary → we vary

L132: oceanfront → ocean front

L157: sea level forcing → sea-level forcing

L169: RMSE → root mean square error (RMSE)

L246: ice sheet configuration → ice-sheet configuration

L254-255: "Additionally, the grounded volume above flotation is 5.3±0.3 m SLE larger, which contributed to the global mean sea-level rise." ← This sentence is ambiguous, should it be rise or reduction? Please modify.

L281: root mean square error (RMSE) → RMSE

L287: surface elevation histories → surface-elevation histories

L287: ice core locations → ice-core locations

L287: in interior Greenland → in the interior of Greenland

L358: the mass loss rates → those

L408: ice flow parameters → ice-flow parameters

L461: ice sheet evolution → ice-sheet evolution

L477: ice sheet simulations → ice-sheet simulations

**Response:** Thank you for the comment.

Thanks for catching these mistakes. We agree that the line about sea-level rise was ambiguous.

> The compound nouns are now hyphenated and the model setup section is in present tense.